
# Hazard Assessment of Earthquake-Induced Landslides Based on a Mechanical Slope Unit Extraction Method, A Case in Ghana

Peter Antwi Buah[1], Yingbin Zhang[1*], Pengcheng Yu[1], Haiying Fu[1], Mingzhe Zhou[1], Qingdong Wang[1], and Jing Liu[1]

[1]Department of Geotechnical Engineering, School of Civil Engineering, Southwest Jiaotong University, Chengdu, China,Sichuan, 611756. P.R. China

**Correspondence:** Yingbin Zhang (yingbinz719@swjtu.edu.cn)

**Abstract.** Slope unit extraction is integral to earthquake-induced landslide analysis. The conventional watershed and hydrological slope unit extraction methods are precarious with a sudden change in slope gradient along the flow direction, which result in slope unit heterogeneity, conjoint slopes, and boundary defects of the extracted slope unit. This paper addresses this research gap by proposing a mechanical slope unit extraction method that combines watershed points, hydrological, and segmentation

methods. This proposed method defines a slope unit as a closed homogeneous space of points overlaid by a mesh having a variance in the slope gradient along its flow direction. The method extracts and uses 3D points to solve slope heterogeneity defects associated with the conventional watershed methods, segmentation to solve boundary defects, and considers the slope pattern and incident ray at a depth to estimate the possibility of earthquake-induced landslides. Ghana (West Africa) is selected to test the proposed slope unit extraction method. The result shows that the method overcame boundary problems, heterogeneity,

sudden gradient change, and conjoint slope unit defects associated with the conventional watershed and hydrological method and shows a uniform slope unit for landslide analysis in Ghana. The landslide prediction rate of Ghana also presents 70.9% landslide inventory, giving an estimated threshold displacement of 9 cm.

Keywords: Slope unit; Earthquake induced-landslide; Mechanical method; Point segmentation; Prediction rate

## 1   Introduction

Earthquakes are the most dangerous natural hazards, posing the most significant risk to life and property. Since the 1980s, earthquakes and the associated landslides have been responsible for nearly half of all-natural disaster deaths (Jibson et al., 2000; Tsai et al., 2019; Osanai et al., 2019; Zhang et al., 2018). Landslides occur on slopes and have been the rationale behind making earthquake-induced landslides and seismic engineering a scientific and national demand. Thus, its evaluation provides general estimates of future earthquake-induced landslides based on medium and long-term predictions of earthquake distribution in

other to provide a possible mitigation measure to curb its impact on life and properties (Bray & Travasarou, 2018; Salunkhe et al., 2017; Tsai & Chien, 2016; Wang & Lin, 2010; Zhang et al., 2019). Historic landslides based on statistical methods were the subject of landslides and slope stability zonation research in the past before advancing into current scientific and engineering stability analysis models (Cencetti & Conversini, 2003; Tsai et al., 2019).



The statistical method could be bivariate methods, such as the frequency method (Chung & Fabbri, 2012; Dai & Lee, 2002; Wubalem, 2020), a multivariate statistical method such as Logistic Regression (LR) method (Atkinson & Massari, 1998; Polykretis et al., 2019), Artificial Neural Network (ANN) method, such as Back-propagation Algorithm method (Ortiz & Martínez-Graña, 2018; Tsangaratos & Benardos, 2014; Vakhshoori et al., 2019) or Machine Learning Techniques (MLTs), such as Support Vector Machine (SVM) method (Tien et al., 2012; Youssef & Pourghasemi, 2021; Kavzoglu et al., 2014). The robustness of the statistical method is, however, suspect. Because the statistical method generates landslide maps by using a combination of maps generated by different control points that are assumed to be conditionally independent of each other, thus questioning its accuracy (Tien et al., 2012; Tsai et al., 2019; Youssef & Pourghasemi, 2021). Recent engineering earthquake-induced landslides and displacement analysis are done using the Newmark's rigid block displacement method, making it the benchmark for contemporary engineering methods (Rathje et al., 1998; Jibson & Keefer, 1993; Saygili, 2008; Tsai et al., 2019; Wang et al., 2017; Shinoda & Miyata, 2017; Zhang et al., 2021). The precision of the Newark Rigid Dynamic Block Model cannot be misconstrued, as it produces a stronger correlation between the estimated sliding block displacement and the mapping location of the earthquake-triggered landslide. This makes the Newark Rigid Dynamic Block Model suitable for the prediction of earthquake-induced landslides (Rathje et al., 1998; Tsai et al., 2019; Xie et al., 2003; Zhang et al., 2019).

The final stage of Newmark's dynamic rigid block model for analysing earthquake-induced landslides and displacement is to generate a landslide hazard map, achievable through slope Mapping units (Schlögel et al., 2018; Yu & Chen, 2020). Selecting an appropriate mapping unit is vital for an efficient landslide susceptibility assessment. Mapping unit for earthquake-induced landslide displacement analysis using Newmark's rigid dynamic block displacement model could either be based on a grid-cell or slope unit model (Schlögel et al., 2018; Tsai et al., 2019; Yu & Chen, 2020). Grid cells are regular square cells with given size for the unit mapping of landslides and are not closely related to geological environments (Guzzetti et al., 1995) . As highlighted by (Xie et al., 2003), a limitation of the grid-cell mapping unit model is its inability to represent natural slopes' topographic boundaries in the real world because it uses artificially marked cells of a block to represent the natural landscape event. According to hydrological theory, a "slope unit" is considered a watershed defined by the ridge and valley lines and is used to divide spaces into minor regions for easy analysis. Slope units are more related to the geological environment, making it the best mapping unit for earthquake-induced landslide and displacement analysis (Ba et al., 2018).The slope unit method for earthquake-induced landslide and displacement analysis is favoured compared to the grid-cell method because landslides occur on slopes, and the slope unit represents topographic features limitations that arise when the grid cell is used (Wang et al., 2017; Xie et al., 2003). Slope unit methods for analyzing earthquake-induced landslides include the curvature watershed method, texture watershed method, standard and inverse-based DEM hydrological method, Conventional Watershed method, Morphological Image Analysis (MIA) methods, and the r slope unit software method e.g. (Alvioli et al., 2016; Cheng & Zhou, 2018; Wang et al., 2019, 2020). The ability of these methods to extract slope unit that reflects the geomorphological features of actual landslides needs verification, because such accurate representation is critical to ensure the physical meaning of subsequent landslide susceptibility analysis (Alvioli et al., 2016; Wang et al., 2019, 2020).

However, these slope unit methods are based on surface hydrological process analysis, making it impossible to identify variations in slope gradient beyond the hydrological flow direction. This results in a sudden change in slope gradient within





a unit, extracting defective slope units (having irregular slope units regions). In this case, the extracted slope units do not

reflect the actual landslides' essential geomorphological features and boundaries. The sudden change in a gradient along the flow direction also causes slope unit heterogeneity, yielding from slope units extracted from high-resolution DEM and mostly happens with slope units extracted using the hydrological method (Guzzetti et al., 1995; Wang et al., 2019, 2020).The conventional watershed method for slope unit extraction also produces irregular parallel boundary and conjoined slope conditions because it barely distinguishes inclined and horizontal planes of deep valleys and high mountainous terrains. Tedious manual

post-extraction corrections are needed to make the slope unit acceptable (Cheng & Zhou, 2018; Wang et al., 2020).

A framework for mechanical extraction of a slope unit using the GIS software is proposed in this paper. The framework combines catchment points, hydrological slope unit extraction method, and segmentation to overcome the limitations of the above slope unit extraction methods. The application of the framework is validated in Ghana. The prediction result of the method is compared with the conventional watershed slope unit extraction and the hydrological method.

The impact of cohesion c is negligible, therefore neglected (Fortt et al., 2007; Matsushi et al., 2006; Yang & Luo, 2015). The paper also underlines the possibility of the proposed model for displacement analysis of shallow and deep slope failures considering the pore water pressure during the computation of the factor of safety$F_s$.

## 2   Proposed slope unit's and displacement method

Considering the slope unit definition and displacement types in Fig.1a, 1b, 1c, and 1d, the flowchart in Fig. 2, is used to predict

the displacement of slope in Ghana (West Africa). The concept predicts the possibility of slope failure under seismic loading during earthquake hazards when the ground vibration exceeds the standard threshold. The earthquake then causes the critical slip surface to move, causing the $F_s$ to fall below one, resulting in an automatic slope collapse and landslide. The displacement method used has been used by many researchers. The $F_s$ becomes the first parameter to be determined after the slope unit is extracted to ascertain the possibility of slope failure (Cheng & Zhou, 2018; Jibson et al., 2000; Tsai et al., 2019; Wang & Lin,

2010). The yield acceleration $k_y$ is then determined to ascertain the rate of displacement. This aspect of the model used for this study has three distinct features compared to the others:

1. A slope unit that accurately predicts the watershed and morphology of the area under consideration by deriving and using 3D points to solve slope heterogeneity defects associated with the conventional watershed methods and segmentation to solve boundary defects.

2. A $F_s$ that considers the pore-water at depth alongside the traditional Newmark's method that neglects cohesion c and considers failure depth d.

3. The $F_s$ and the $k_y$ are all computed for using ArcGIS to eliminate common iterative errors.




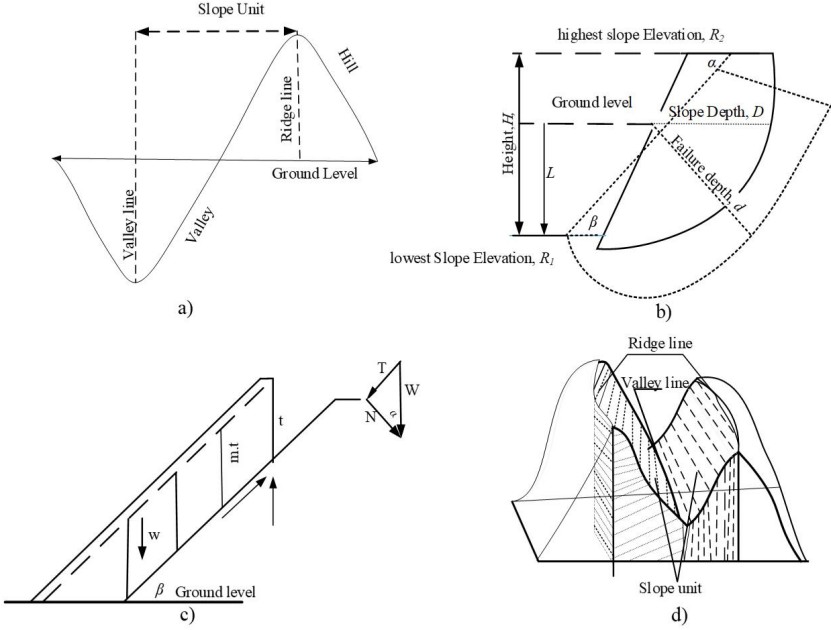

**Figure 1.** Slope Unit and sliding types a) 2D Slope Unit, indicating ridge and valley lines. b) 2D view of circular slope failure model under static and failure mode, where $\beta$ is the slope angle with ground and $\alpha$ is the slope angle of failure. c) 2D View of Slope in Plane Failure Mode, where $W$ is the weight of soil mass, $L$ is the length and t the depth of the slope. d) Slope unit in 3D view showing ridge and valley lines.

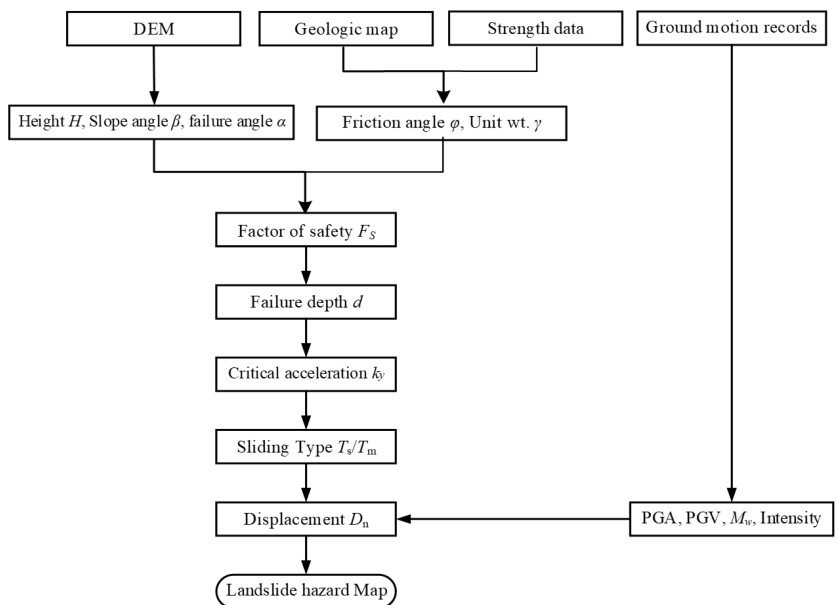

**Figure 2.** Flowchart for the proposed displacement prediction for slope units.



.

Step 1. Slope unit: A slope unit is an area's geomorphology, segmented into smaller mapping units by ridge and valley lines

or the left or right sides of a watershed's sub-basin Fig.1a and d. Slope unit plays an essential role in the prediction of slope displacement landslide (Cheng & Zhou, 2018). A combination of Snap points, hydrological watershed, and segmentation methods (Cheng & Zhou, 2018; Ho & Gibbins, 2009; Wang et al., 2017) is preferred. This slope unit extraction method operates on an algorithm in which the watershed is based on a DEM in a 3D form, implying that the DEM terrain is in 3D points overlaid by mesh in the longitudinal and vertical directions (Ho & Gibbins, 2009). The Variance of the watershed

3D is applied to the DEM imagery gradient to solve the DEM boundary, change in gradient, and conjoint slope problems associated with slope units extracted by Hydrological and conventional watershed methods. The method involves a catchment basin divided by watershed lines using the flowchart in Fig. 3. The steps involve first demarcating pour points to indicate the lowest point of the surface where water flows out of the catchment. The sink is then determined to reveal the DEM problems before filling to rectify them. Flow direction and accumulation are extracted to aid in the catchment watershed extraction. Snap

pour points are then delineated to snap the pour point to the closest highest accumulation cell. A watershed is obtained. The procedure is repeated with an inverted DEM. The mountain and crevasse watersheds obtained are merged and segmented to delineate the boundaries before the morphological ridge and valley lines are determined to end the slope unit extraction.

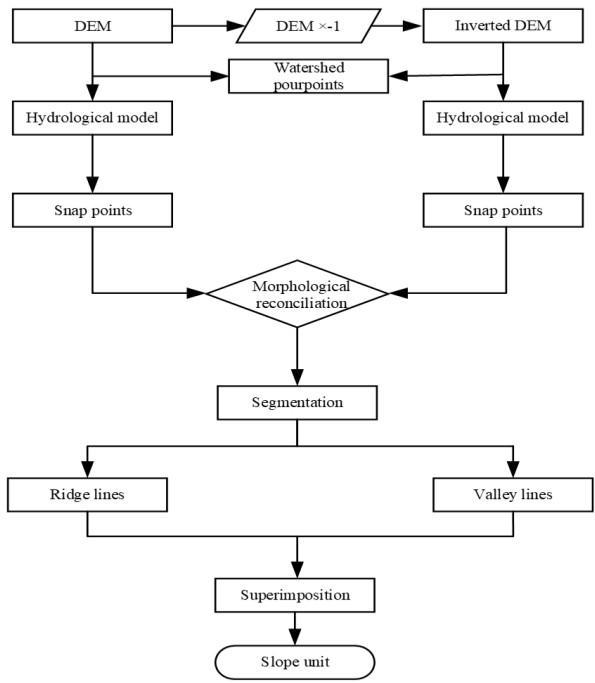

**Figure 3.** Flowchart for slope unit extraction procedure.

.




Step 2. Slope angle: The slope angle is the ratio of the riser to the run. In determining the slope unit, the slope's height $H$
and angle $\beta$ are determined manually using Eq. (1) and Fig.4 or with GIS software Fig. 8 (b).

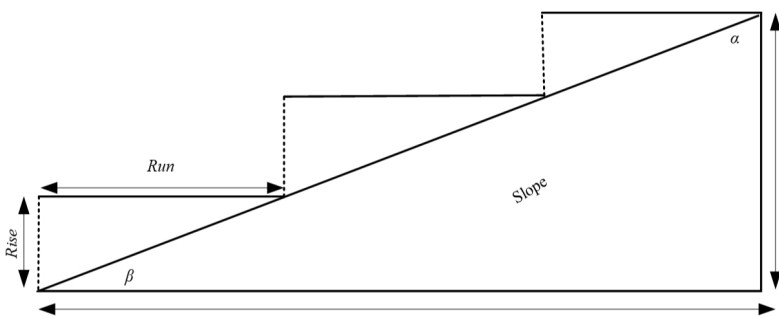

**Figure 4.** Slope angle indicating a Rise and Run.

$$\beta = \tan^{-1}\left(\frac{rise}{run}\right) \tag{1}$$

Step 3. Factor of safety $F_s$: $F_s$ is the ratio of a surface's shear stress ($\tau_{ss}$) to the available stability strength ($\tau_{st}$) that prevents
the body from collapsing. The $F_s$ is used as a measure for determining slope's stability. A static yet near-unstable slope has a
factor of safety below or equal to one. A $F_s$ less than one indicates an unstable slope that has failed or likely to fail (Salunkhe
et al., 2017; Tsai & Chien, 2016), whereas a $F_s$ above 1.5 indicates a stable slope (Salunkhe et al., 2017; Tsai & Chien, 2016).
This analogy of an $F_s$ makes its determination extremely important, especially when the slope's cohesive strength is ignored,
leaving the slope weak and exposed. This supports the analogy for designing against the worst case, which is noted to be the
best engineering practice according to Taylor's stability (Sahoo & Shukla, 2019). Eqs. (2-6) sums up the $F_s$ method used in
this study.

$$F_s = \frac{\tau_{ss}}{\tau_{st}} = \frac{c + \sigma \tan\varphi}{\tau} \tag{2}$$

where $c$ is the cohesion, $\sigma$ is the effective stress, $\tau$ is tensile stress, and $\phi$ is the rock's friction. $\sigma$ can be derived by Eq. (3).

$$\sigma = (\gamma - \gamma_w m)\, d\cos\beta \tag{3}$$

where $\gamma$ is the dry unit weight of soil, $\gamma_w$ is the wet unit weight, $d$ is failure depth, $\beta$ is the failure slope angle and $\tau = \gamma d\sin\beta$,
the Factor of safety $F_s$ could then be written as in Eq. (4).

$$F_s = \frac{c + [(\gamma - m\gamma_w)\, d\cos\beta]\tan\varphi}{\gamma d\sin\beta} \tag{4}$$

Where m is the percentage of failure thickness saturated. This study neglects the effect of cohesion in its analysis to generate
the $F_s$ map in Fig. 7 (a). The $F_s$ is therefore written as in Eq.(5).





$$F_s = \frac{(\gamma - m\gamma_w)\,d\cos\beta\tan\varphi}{\gamma d\sin\beta} \tag{5}$$

Eq. (5) is primarily suitable for an infinite slope, because the $F_s$ for infinite slopes is not dependent on the slope's depth $d$ but rather the $\phi$ and $\beta$. The approach could also be used to compute for the $F_s$ of a finite slope by adding a depth correction factor ($DCF$) since with finite slope failure, unlike the infinite, depends on slope depth d Eq.(6).

$$F_s = \frac{(\gamma - m\gamma_w)\,d\cos\beta\tan\varphi}{\gamma\,d\sin\beta} r_u \tag{6}$$

where $r_u$ is the pore water pressure distribution (Sun & Zhao, 2013) distressing the shear stability at a depth per unit area of the slope, affecting its stability in Eq.(7).

$$r_u = \frac{u}{\gamma d} \tag{7}$$

where $u$ is the pore water pressure, and d is the slope failure depth, ru could also be considered as the ratio of underground water to the slope's height (Sun & Zhao, 2013). Circular displacement, D/H could be obtain from the DEM and chosen based on the direction of the force acting on the slope, using Eq.(8) (Saygili & Rathje, 2009).

$$\frac{D}{H} = \left[\frac{0.16}{\tan\beta} + 0.081\right] + \left[\frac{-2.35}{\tan\beta} + 3.77\right] \times \left[\frac{c/\gamma H}{F_S}\right] + \left[\frac{42.0}{\tan\beta} - 35.2\right] \times \left[\frac{c/\gamma H}{F_S}\right]^2 \tag{8}$$

$$\alpha = \tan^{-1}\left[\frac{H}{R_1 + R_2 + \cot\beta H}\right] \tag{9}$$

Where $R_1$ and $R_2$ are the lowest and highest slope Elevations, respectively, in Fig. 1 (b), $D$ is the soil thickness, $H$ is the slope height, and $\phi$ is the soils frictional angle, which is dependent on the soil or rock type (Dunne et al., 2011; Terzaghi et al., 1996) (Dunn et al., 2011; Terzaghi et al., 1996). Thus, in most cases, the failure depth d depends on the soil thickness $D$, which may be minimal (in some circumstances) due to a higher slope angle, $\beta$ indicating that a very high slope angle depicts a possible plane failure circular.

Step 4. Yield acceleration, $k_y$(g): Slope properties are influenced mainly by their yield acceleration properties, including groundwater level, geometry, and material strength. The $k_y$(g) of the slope is defined as a sliding that commences when vibrating acceleration exceeds its threshold causing block of slopes held in place to move along a sloppy surface until the relative velocity between the block and the ground is zero as in Fig. 7 (b) using Eq.(10).

$$k_y(g) = ((F_S - 1)\,g \cdot \sin\beta\,(D_{CF} + 1)) \tag{10}$$


Assuming force direction is parallel, where m is the moment magnitude and the $DCF$ is the depth correction factor Eq.(11).

$$D_{CF} = \begin{cases} \exp\left(0.4 + 0.343\tan\varphi\frac{D}{H} - 1.5\frac{D}{H}\right) & (\beta - \alpha \geq 5) \\ 0 & (\beta - \alpha < 5) \end{cases} \tag{11}$$

where $\alpha$ is the failure angle.

Step 5. Sliding type, $T_s/T_m$: The downward movement of soil mass in a block avalanche due to seismic activities, especially earthquakes, is termed displacement. Its determination is crucial in slope stability and displacement analysis because it indicates the slope's behavior. Displacement could be computed for as a rigid block or a flexible sliding body. The sliding type is computed as rigid or flexible by finding the relationship between the mean slope period Ts, and the ground motion spectral acceleration Tm (Rathje et al., 1998). $T_s/T_m$ expressed as in Eq. (12).

$$\begin{cases} rigid\,block & \frac{T_s}{T_m} < 0.1 \\ flexible\,block & \frac{T_s}{T_m} > 0.1 \end{cases} \tag{12}$$

where $T_m$ is the ground motion spectral acceleration at a degraded period of the slope ($1.5T_s$) (Bray & Travasarou, 2018). $T_s$ is the mean initial acceleration period of the slope.

$$T_S = \frac{4H}{V_S} \tag{13}$$

where $H$ is height or depth of sliding block and $V_s$ is the shear Wave velocity of slope (usually $V_{s,30}$ = 760 m/s) for rock
site conditions (Bray & Travasarou, 2007). $T_m$ can also be evaluated from the Fourier amplitude spectrum and is defined as in Eq.(14) (Du & Wang, 2017; Zhang et al., 2019).

$$T_m = \left(\frac{\sum_i \left(c_i^2/f_i\right)}{\sum_i c_i^2}\right) \tag{14}$$

where $C_i$ is the Fourier amplitude coefficients of the ground motion (gm) at frequency $0.25 \leq f_i \leq 20$ Hz, and $f_i$ is the discrete Fast Fourier Transform, $FFT$, frequencies. Eq.(14) is reliable for an earthquake magnitude range of 3 to 7.9 and
rapture distance up to 300 meters, showing a direct relationship between $T_m$ and moment magnitudes of ground motion ($gm$). $Mt > 7.0$ always provides reasonable sequential results, and because Ghana has no ($gm$) above 7.0, $T_m$ of 0.82 s is acceptable.

Step 6. Rigid Block Displacement, ($D_n$): The Rigid block Displacement model proposed by (Zhang et al., 2019) is preferred due to its lower error rate and higher efficiency than other displacement models. The rigid block displacement is dependent
on peak ground velocity ($PGV$) and peak ground acceleration ($PGA$) values since the period ($T_s$) of a sliding mass is null, the dynamic response could be insignificant (Rathje & Antonakos, 2010). Therefore, this method, which has all parameters centered on the ($PGV$) and ($PGA$) of the ground motion, is deemed appropriate because the research assumes a slope failure could only occur when a ground motion exceeds the slopes' resistance strength in the research area (Ghana).

$$\log D_n = A\log\left(1 - k_y/PGA\right) + B\log k_y + C\log\left(PGV\right) + D \pm \varepsilon \tag{15}$$




where $A, B, C$ and $D$ are coefficients and $D_n$ is the displacement (in, cm), $k_y$ is yield acceleration due to gravity, $\epsilon$ is the standard deviation of the model with zero mean. PGA is the peak ground acceleration (g), and $PGV$ is the peak ground velocity (in cm/s).The model is then written as in Eq. (16).

$$\log D_n = 2.47 \log (1 - k_y/PGA) - 0.917 \log k_y + 1.480 \log (PGV) + 2.027 \pm 0.366 \tag{16}$$

Step 7. Flexible Block Displacement, $(D_n)$: Flexible displacement of slope could be determined using peak ground accel-
eration, mean periods, seismic coefficient, seismic coefficient time history, natural period of sliding mass and mean period of the earthquake ($PGA, T_m, k_{max}, k_{velmax}, T_s, T_m$). where the $PGA$ and $PGV$ for shallow failure translates into $k_{max}$ and $k_{velmax}$ as was proposed by (Rathje & Antonakos, 2011; Tsai & Chien, 2016), and the dynamic response of flexible sliding block interacts with incident motion is expressed as in Eq.(17).

$$for\ [T_s/T_m \geq 0.1]$$
$$\ln (k_{\max}/PGA) = \left\{ \left[ (0.45 - 0.702 PGA) . \left( \tfrac{\ln(T_s/T_m)}{0.1} \right) \right] + \left[ (-0.228 + 0.076 . PGA) . \left( \tfrac{\ln(T_s/T_m)}{0.1} \right)^2 \right] \right\}$$
$$for\ [T_s/T_m \leq 0.1]$$
$$\ln (k_{\max}/PGA)\ = 0 \tag{17}$$

$$for\ [T_s/T_m \geq 0.2]$$
$$\ln (k_{\max}/PGA) = \left\{ \left[ (0.24) . \left( \tfrac{\ln(T_s/T_m)}{0.2} \right) \right] + \left[ (-0.091 - 0.171 . PGA) . \left( \tfrac{\ln(T_s/T_m)}{0.2} \right)^2 \right] \right\}$$
$$for\ [T_s/T_m \leq 0.2]$$
$$\ln (k_{\max}/PGA)\ = 0 \tag{18}$$

This model is selected because it can be applied to more profound flexible failures and largely depends on the sliding mass period (Tsai & Chien, 2016).

## 3 Case Study - Ghana (West Africa)

The study area is Ghana (West Africa), bounded by the four cardinal coordinates $11.054^o$N $0.285^o$W (northeast), $6.1256^o$N
$1.2254^o$E (southeast), $4.866^o$N $2.24090^o$W (Southwest), $10.8783^o$N $1.9833^o$W (Northwest). Ghana's terrain is characterized by small desert mountains in Kwahu, Mampong, Akuapim, and Afajato. The highest elevation is $885_m$ ($2,904_{ft}$.) above sea level at the southwestern part (Volta region) and other steep valleys. Ghana is tropical and experiences a rainfall range between 78 to 216 centimeters (31 to 85 inches) per year, from April to mid-November. Ghana has predominantly flat land in some parts with no slope, while other areas with slopes are steep slopes above $45^o$, Fig. 8 (b),appendix.1.

### 3.1 Seismic Activities in Ghana

Landslides have claimed many lives and properties worldwide over the years. They occur when gravity overcomes frictional forces, keeping layers of rocks and soils in place. As a global menace, Landslide has occurred in almost every part of the world,


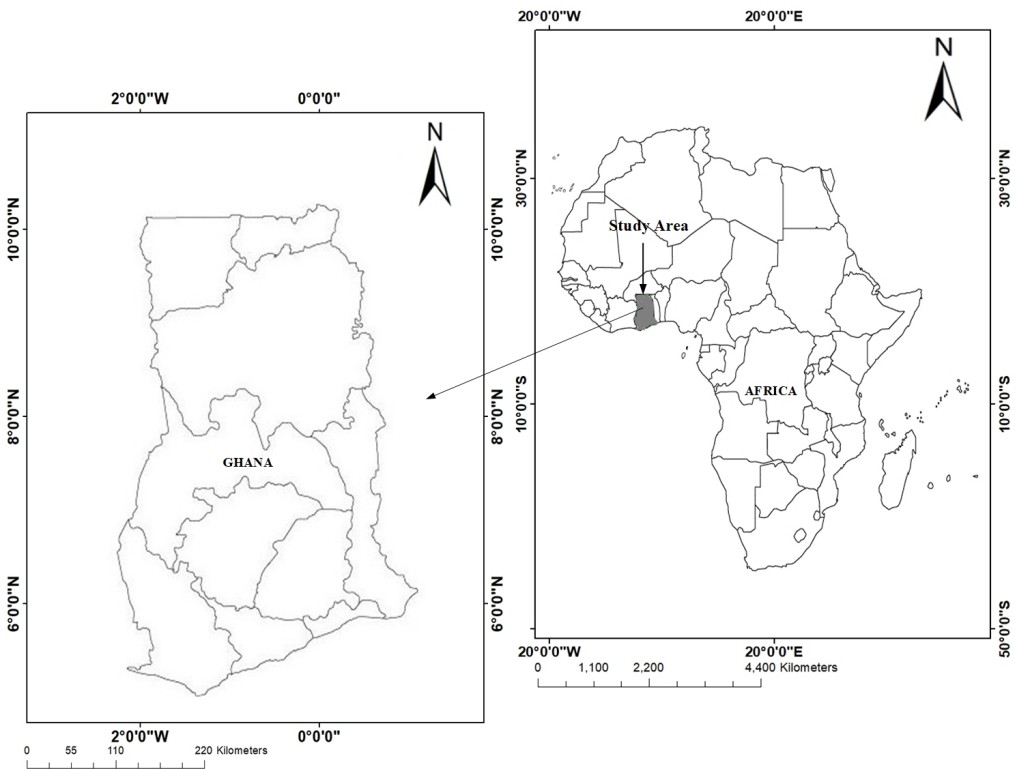

**Figure 5.** Location of Ghana in Africa.

including West Africa (Salvador, 2016). The 2017 Sierra Leone landslide (mud-flow) took more than 400 lives and rendered 3000 people homeless; the Democratic Republic of Congo (DRC) Landslide killed at least 200 people and covered millions
of dollars of resources. Nigeria, Congo, Cameron, Uganda, and Ghana have also been affected by these landslides, mainly due to rainfalls and seismic activities (Kervyn et al., 2016). Ghana (West Africa) is hereby selected to implement the new slope unit extraction method. Ghana has had some minor landslide cases in the past. These landslides occurred due to rainfall and earth tremors (Ghana Institute of Geoscientists, GhIG,). With regards to seismic, Ghana recorded its first earthquake in 1615 and all subsequent ones in Table. 1 (Amponsah et al., 2009). The earthquakes in Ghana are caused by the continuous
strike-slip movement on the "Romanche Fracture Zone" adjacent to the West Africa continental margin (Blundell et al., 1976). The western part of Accra (weija) on the junction of the coastal boundary and the Akuapim fault, has experienced most of the earthquakes in Ghana, making it the epicenter of earthquakes (Bates, 1962). Ghana's landscape has low and high lands, with a rainfall pattern for a minimum of five months per annul. Some periodic earthquakes record has forced GhIG to predict the likelihood of a possible landslide in Ghana. The maximum intensity of the earthquake in Ghana is measured to be IX on the
MSK Scale, recorded in 1862 (Ambraseys & Adams, 1986). The highest Peak Ground Acceleration (PGA) recorded in Ghana was at the Accra-Tema seismic zone with an estimation of about 0.2g and minimized to 0.05g with 140 km away from Accra





(Stevens et al. 2018). Ghana's Peak Ground Velocity (PGV) ranges from 9.2 to 37.1 cms-1, and the standard PGA ranges from 0.14 to 0.2g (Amponsah et al., 2009). The areas with low PGV in Ghana usually display high PGA (Amponsah et al., 2009).

**Table 1.** Earthquake Record of Ghana indicating earthquake parameters

| No. | Year | Magnitude ($Ml$) | Intensity ($In$) | Surface Magnitude ($Ms$) | Source |
|---|---|---|---|---|---|
| 1 | 1615 | | | | Ambrasey's and Adaams, 1986 , *NNA* |
| 2 | 1636 | 5.7 | IX | North Axim | Ambrasey's and Adaams, 1986, *NNA* |
| 3 | 1788 | 5.6 | | Accra | British Geological Survey, *BGS* |
| 4 | 1862 | 6.8 | IX | Accra | Ambrasey's and Adaams, 1986, *NNA* |
| 5 | 1858 | 4.5 | | West of Accra | Ambrasey's and Adaams, 1986, *NNA* |
| 6 | 1871 | 4.6 | VI | Accra | Ambrasey's and Adaams, 1986, *NNA* |
| 7 | 1872 | 4.9 | VII | Accra | Ambrasey's and Adaams, 1986, *NNA* |
| 8 | 1879 | 5.7 | | Accra | Ambrasey's and Adaams, 1986, *NNA* |
| 9 | 1906 | 6.2 | VIII | Ho | Ambrasey's and Adaams, 1986, *NNA* |
| 10 | 1907 | 5 | | Accra | Ambrasey's and Adaams, 1986, *NNA* |
| 11 | 1939 | 6.5 | IX | Accra | Ambrasey's and Adaams, 1986, *NNA* |
| 12 | 1948 | 4 | | Accra | Ambrasey's and Adaams, 1986, *NNA* |
| 13 | 1964 | 4.7 | | Near Akosombo | Akoto and Annum, 1992, *AKO* |
| 14 | 1969 | 4.8 | | Offshore | United State Geological Survey, *USG* |
| 15 | 1997 | 4.7 | | Accra District | Internal Seismological centre, *ISC* |
| 16 | 1992 - 2002 | 3-Jan | IV | Ho, Accra | Ambrasey's and Adaams, 1986, *NNA* |
| 17 | 1615 | 4 | | Accra | Ambrasey's and Adaams, 1986, *NNA* |

## 3.2 Geologic formation of Ghana

Previously known as "Gold Coast," Ghana got its name from its rock units (gold deposits) formed some one billion years or older. Two-thirds of Ghana's Land area is covered by the Birimian Rock of Neoproterozoic age (covering Northern, central, and western) and are mainly sedimentary formed from volcanic rock sediments (Hirdes et al., 1992). The Proterozoic rock unit covers the remaining one–third and at Dohomanya, Togo, Buem, and the Voltarian belt (Leube et al., 1990). The protozoic rocks comprise mainly of igneous crystallization ages of four granitoid formed some 500 years ago (Hirdes et al., 1992).
Crystalline basement rock (West African Shield), volcanic belt rocks, sedimentary basins affected by igneous activities, and two significant mountain orogeny are the main geologic formations of Ghana, Fig.8 (a).

## 3.3 Research data

The data used for this research are the DEM of Ghana (available at a 60 m × 60 m pixel resolution and was re-sampled to a 30 m×30 m pixel resolution), Geologic Map, and Strength Data. The DEM was used to obtain the Topographic Position


**Table 2.** Major Geologic formations in Ghana

| Land Area 92,099$^2$ (miles) | % | Rock Type | Rock Components | Location |
|---|---|---|---|---|
| 61,399.32 | 2/3 | Birimian Rock | Metamorphosed volcanic sedimentary, plutonic alkaline and granites rocks | North, West, and Ashanti |
| 30,699.62 | 3-Jan | Protozoic Rock | Granitoids of igneous rocks | East, Volta, Accra |

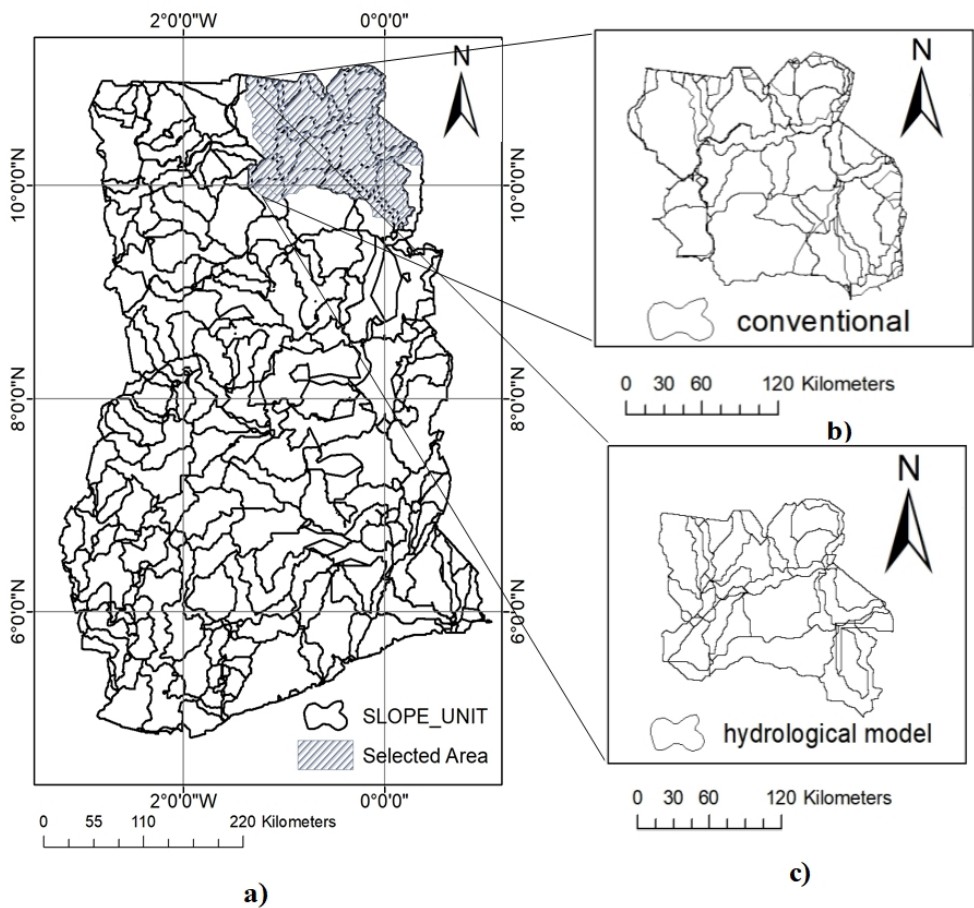

**Figure 6.** Slope Units of Ghana obtained using different methods a) Snap point segmentation b) Conventional watershed c) Hydrologic

Index ($TPI$), slope gradient, elevation, and other topographic parameters. The geological information was provided by the Geological Survey Department of Ghana on a 1:100,000 scale map. Strength data was derived from the geological features and other information available, including triaxial tests, as in the case of (Jibson et al., 2000). The required strength parameters used include the $\gamma$ and $\phi$.





**Table 3.** Location of the Major Geologic Unit in Ghana

| No | Unit | Abbreviation | Rock Components | Location |
|----|------|--------------|-----------------|----------|
| 1 | Precambrian | PC | Volcanic plutonic alkaline | Kumasi, sefwi, brong, central, wassa, some part of the north |
| 2 | Ordovician - Cambrian | OCM | Sedimentary Mudstone and Siltstone | Volta Basin |
| 3 | Water | H20 | Water (some gold rocks) | Volta lake, Kumasi, tarkwa |
| 4 | Holocene | QE | Granatoids of igneous rocks | Accra |
| 5 | Tertiary | T | Quazite, shales Granites of Igneous Rocks | Enchi and its environs. |
| 6 | Quaternary Tertiary | QT | sandstones | Kwahu volta |
| 7 | Carboniferous Devonian | CD | Quartz and conglomerates | Takoradi, secondi, axim |

**Table 4.** Strength Data indicating friction angles and unit weight for the various Geologic Units in Ghana

| Rock Strength Parameters | Frictional Angle $\phi(^o)$ medium | High $\phi(^o)$ | Low $\phi(o)$ | Unit weight $(\gamma)$ kN/m³ medium | High $(\gamma)$ | Low $(\gamma)$ | Source |
|--------------------------|-----------|------|-----|------|------|------|--------|
| Volcanic plutonic alkaline | 50 | 55 | 40 | 26 | 30 | 20 | Goodman RE 1974, 1976 |
| Sedimentary Mud-stone and Silt-stone | 45 | 50 | 40 | 25 | 30 | 20 | Goodman RE 1974, 1976 |
| Granatoids of igneous rocks | 45 | 50 | 40 | 28 | 30 | 24 | Goodman RE 1974, 1976 |
| Quazite, shales Granites of Igneous Rocks | 45 | 50 | 40 | 28 | 30 | 24 | Goodman RE 1974, 1976 |
| Sand-stones | 30 | 34 | 27 | 24 | 28 | 20 | Goodman RE 1974, 1976 |
| Sedimentary Mud and Silt stone | 34 | 39 | 29 | 18 | 23 | 14 | Goodman RE 1974, 1976 |

## 3.4 Probability of Slope Failure

The $F_s$ is a measure of the possibility of failure of an area under study. "$F_s \leq 1$" indicates the likelihood of failure, whiles "$F_s \geq 1.5$" is an indication of stability (Fig.7(a); Schroeder & Swanston, 1987). The Poisson process, using (Wang et al., 2014) can also be used to indicate the possibility of failure and works just like the $F_s$. In Eq. (19), the Poisson failure principle is used in this research to verify the accuracy of the $F_s$.

$$P[E] = 1 - \exp(1 - \lambda.T) \tag{19}$$

where $P$ is the probability of a Failure, $\lambda$ is the rate of occurrence, $T$ is the event's time interval.

## 3.5 Failure Seismic Magnitude

It is necessary to determine the magnitude of seismic excitation that can trigger slope displacement, Eq.(20) is used to determine the extent of earthquake vibration that can trigger slope displacement in Ghana (Jibson & Keefer, 1993).

$$\log I_a = M_w - 2 \log R - 4.1 \tag{20}$$

Where $I_a$ is the earthquake arias intensity, $M_w$ is the moment magnitude, and $R$ is the distance from slope to earthquake epicentre (in kilometers).





**Table 5.** Source of Data used for the research

| No. | Data Title | Source | Format | Usable Fit |
|-----|-----------|--------|--------|-----------|
| 1 | DEM | USGS | Raster | Raster |
| 2 | Strength Data (Geological Map) | Ghana Geo-technical Department | PDF | ArcMap |
| 3 | Ground Motion Records | Ghana | PDF | PDF |

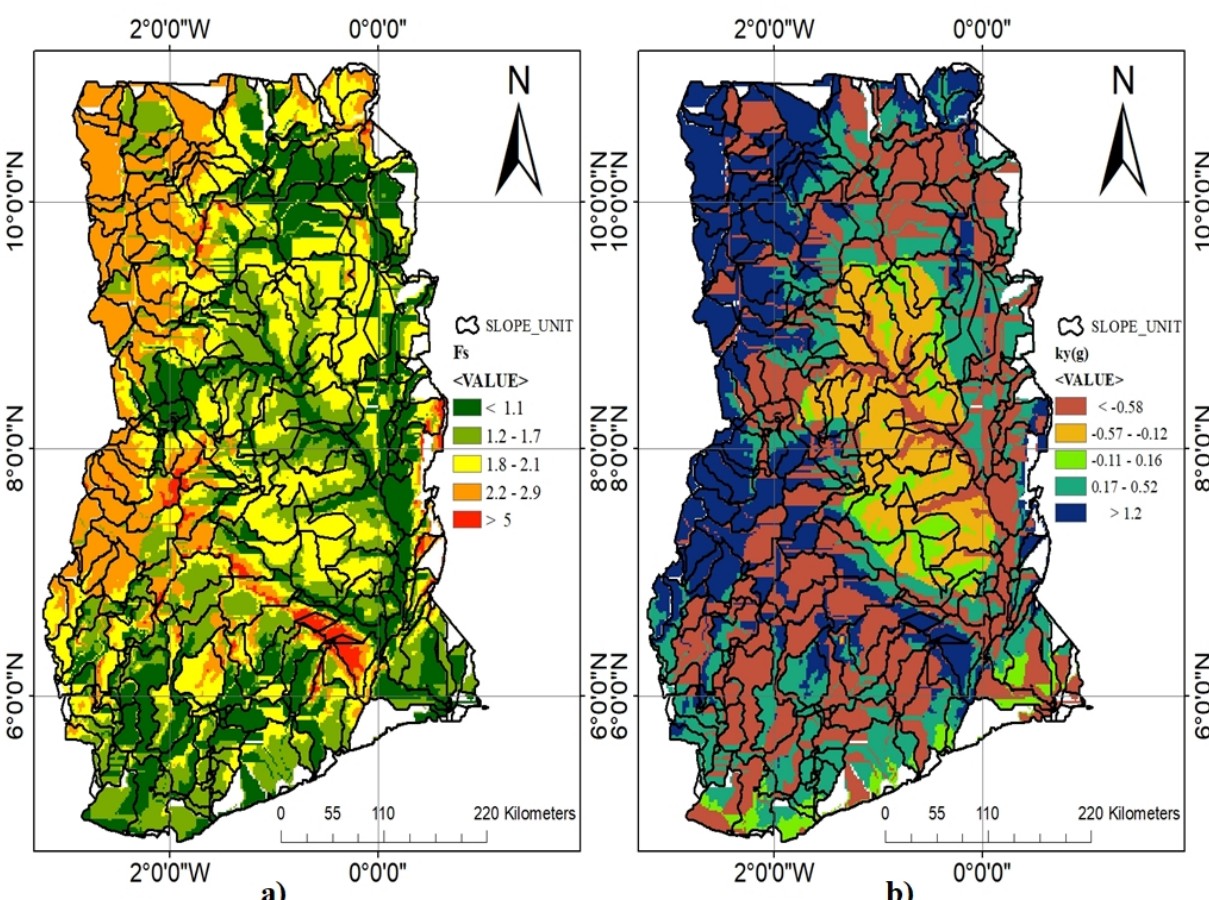

**Figure 7.** Automatic generated Maps a) Factor of Safety, $F_s$, b) Yield acceleration, $k_y$,

## 4 Implementation of the proposed method

The seismic loading of Ghana is done considering a PGA of 0.13g, $PGV$ = 23.1 cm/s, maximum input $PGA$ (Kmax = 0.2g), $PGV$, ($k_{velmax}$=30 cm/s), $T_{mmax}$= 0.82 s and $T_s$ = 0.4 s to compute for the deterministic rigid and flexible displacement of

Ghana.



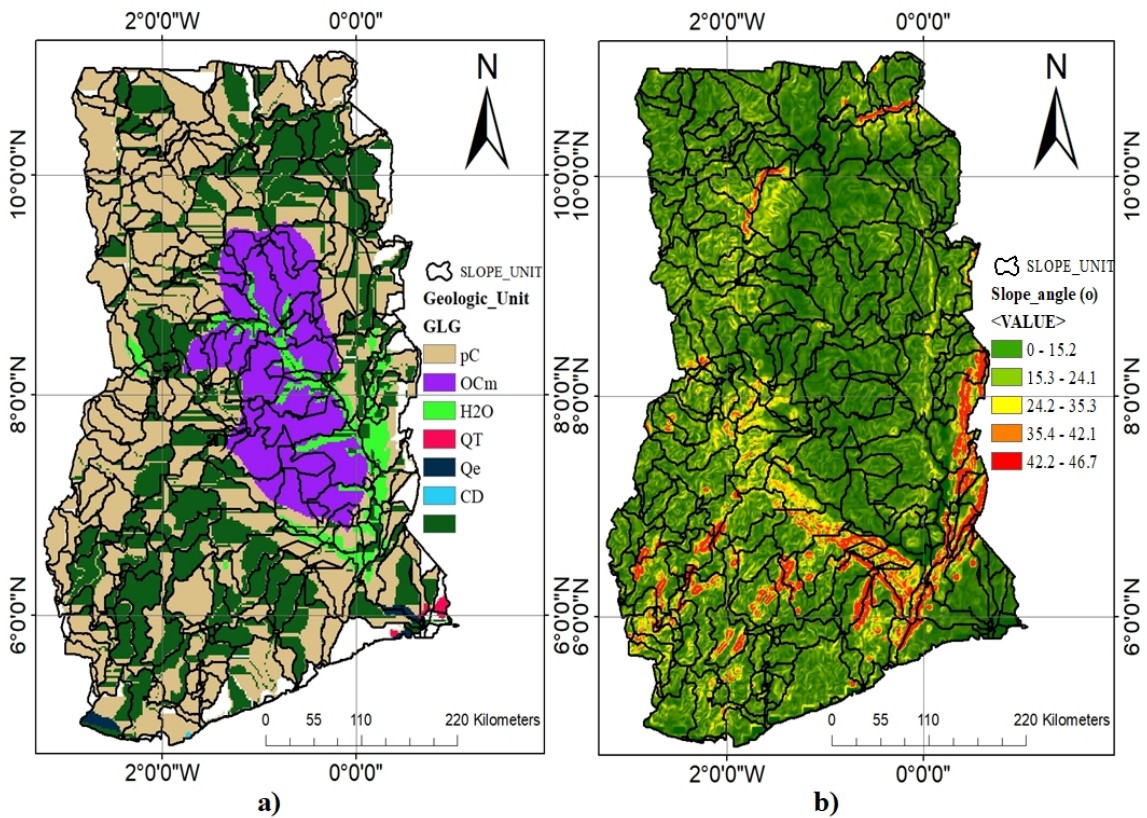

**Figure 8.** Major Lithostratigraphic complex of Ghana a) Geologic Unit b) Slope Angles

## 4.1 Slope Unit, Factor of safety, $F_s$, and Critical acceleration, $k_y$

The primary step towards the actualization of a detailed displacement analysis is to first extract the slope unit. A slope unit is the geomorphology of an area with a divide, obtained using a digital elevation model (DEM). Slope units also divide vast areas (study area) into smaller watershed blocks to aid easy analysis. A combination of the pour points, inverse hydraulic, and segmentation method is used alongside Archydro tools in ArcGIS to extract the slope unit in this study (Cheng & Zhou, 2018; Akagunduz et al., 2008; Wang et al., 2017). The flowchart in Fig.3 is used to guide the slope unit's extraction for this study Fig.6 (a). The method is selected for its precision and an overhaul of the standard hydrological slope unit extraction method in Fig.6 (c) and the conventional watershed method Fig.6 (b) and could extract a flawless slope unit. The $F_s$ is computed using Eq. (6) and the strength parameters in Table.4 in ArcGIS software to obtain the $F_s$ in Fig.7 (a). Locations with deep green colors have low $F_s$ ($F_s < 1$) hence unstable, while Locations with red, yellow and light green colours have high $F_s$ ($F_s > 1$) (Salunkhe et al., 2017; Tsai & Chien, 2016). The $k_y$(g) is automatically generated in ArcGIS software using Eqs. (10 and 11).


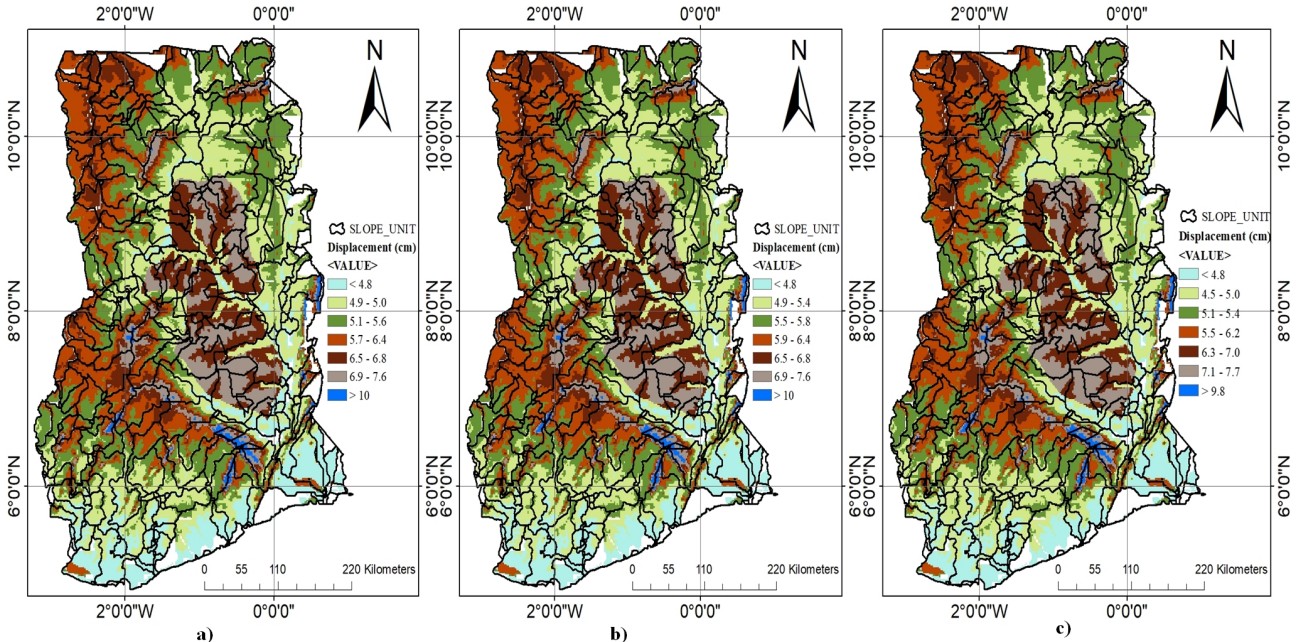

**Figure 9.** Displacement maps: a) (Tsai & Chien, 2016); b) (Jibson et al., 2000)Jibson et al. (2000); c) (Zhang et al., 2019)Zhang et al. (2019)

Considering the $k_y(g)$ map of Ghana in Fig.7 (b), lower $k_y(g)$ values are at areas with brown colors. In comparison, the areas with low and high $k_y(g)$ values Fig.7 (b) correspond to low and high $F_s$ values Fig.7 (a).

## 4.2  Displacement

The sliding type determination is vital, it indicates whether a rigid or flexible sliding mass is likely to occur. Eq.(12) is used to determine the sliding type in this research, where $T_s/T_m$ below 0.1 indicates rigid failure and above 0.1 indicates flexible failure, (Rathje et al., 1998). The displacement in Eqs. (16 and 17) are selected to predict the displacement type as flexible or rigid. Fig.10(b) is the displacement map of Ghana indicating all probable failed areas generated using ArcGIS software. Probable failure areas in the country are Tutukpene, Ho, Hohoe (Volta Region), Somanya, Adukrom, Kwahu, Tafo, Adawso,

Eburi (East Region), Asikuma (Central Region), Enchi (Western North Region), Bekwai Ahwainso, Sefwi Wiawso (Western North Region), Dunkwa (Central Region), Jasikan (Oti Region), Nkwanta (Oti Region), kintampo (Bono East Region), Daboya (Savanna Region), Gambaga (North East), and Wale wale (North East), according to the displacement map. To determine the earthquake's magnitude to cause slope displacement in Ghana, Eq. (20) is used. IX earthquake intensity (highest ever recorded in Ghana) and 80 kilometer from the epicentre (weija) is selected. This research can confirm that an earthquake

with a magnitude above 7.9 could trigger slope displacement in Ghana. The slope failure is determined using the prediction accuracy method by (Jibson & Keefer, 1993; Tsai et al., 2019). The method states that the prediction accuracy for slope displacement under seismic loading should depend on the relationship between the threshold and predicted displacements,





noting an allowable threshold displacement between 5 - 10 cm depending on the residual slope. The worst case of slope units
with displacement above 10 cm was selected as the failed areas in this study using the suggestion by (Jibson & Keefer, 1993)

as the basis for the selection. The failed slope units selected were consistent with the Landslide susceptibility map of Ghana
in Fig.10 (a) obtained using the Frequency Ratio ($F_r$) method by (Bu et al., 2019). Fig.11 (a) indicates a 9 cm threshold
displacement for slopes in Ghana, which means any slope units with a displacement Greater than 9 cm ($D_n > 9$) is a failed or
liable to fail slope.

## 4.3 Accuracy of the displacement prediction

To emphasize the quality of the displacement model used in this study, the pixel (Prediction Rate) method by (Wang & Lin,
2010) is used.The pixel procedure (Prediction rate) introduced by (Wang & Lin, 2010) is used in this study by representing
the pixels with slope units in this study. The prediction rate is represented by $P_r$, which is a ratio of the slope units, with
accurate predictions to the total number of slope units. The accurate predictions encompass slope units containing failure scars
and with displacement greater than threshold displacement ($S_1$) and slope units without failure scars and having displacement

lower than the threshold displacement ($S_4$). The total number of slope units also includes the incorrect predictions, which also
encompasses slope units without scars but displacement above the threshold ($S_2$). And those with scars and displacement are
less than the threshold ($S_3$) Table.6. For this Method, ($P_r$) is 53%, with 127 accurate predictions out of 241 slope units at a
threshold displacement of 5 cm. The Method also presents a 70.9% Pr with 171 accurate predictions out of 241 slope units at a
threshold displacement of 10 cm. Per Fig.11 (a), the threshold displacement of slopes in Ghana is 9 cm, which agrees with the

suggestion by (Jibson & Keefer, 1993).

$$P_r = \frac{S_1 + S_4}{S_1 + S_2 + S_3 + S_4} \tag{21}$$

**Table 6.** Displacement prediction parameters S1, S2, S3 and S4

| No. | Slope Unit Location identification | Symbol | Accurate Prediction | Inaccurate Prediction | Values at 5 cm displacement | Values at 10 cm displacement |
|---|---|---|---|---|---|---|
| 1 | With scars and displacement above threshold | S1 | S1 | | 73 | 73 |
| 2 | Without Scar and displacement below threshold | S4 | S4 | | 54 | 98 |
| 3 | Without scars and displacement above threshold | S2 | | S2 | 114 | 70 |
| 4 | With scars and displacement less than threshold | S3 | | S3 | 0 | 0 |

## 4.4 Failure Rate

According to (Tsai et al., 2019), a predicted slope failure does not necessarily indicate that the entire slope within a slope unit
will fail but rather a portion. Therefore, the percentage of predicted slope failure in Ghana is computed using the ratio of failure

within the slope unit, $u$, to the slope unit's entire area, $a$, Eq.(22). The analysis indicates that the predicted failure rate of slope
in Ghana is likely to be between 11.3 to 24 percent. This shows that Ghana's possible landslide hazard wouldn't be higher than

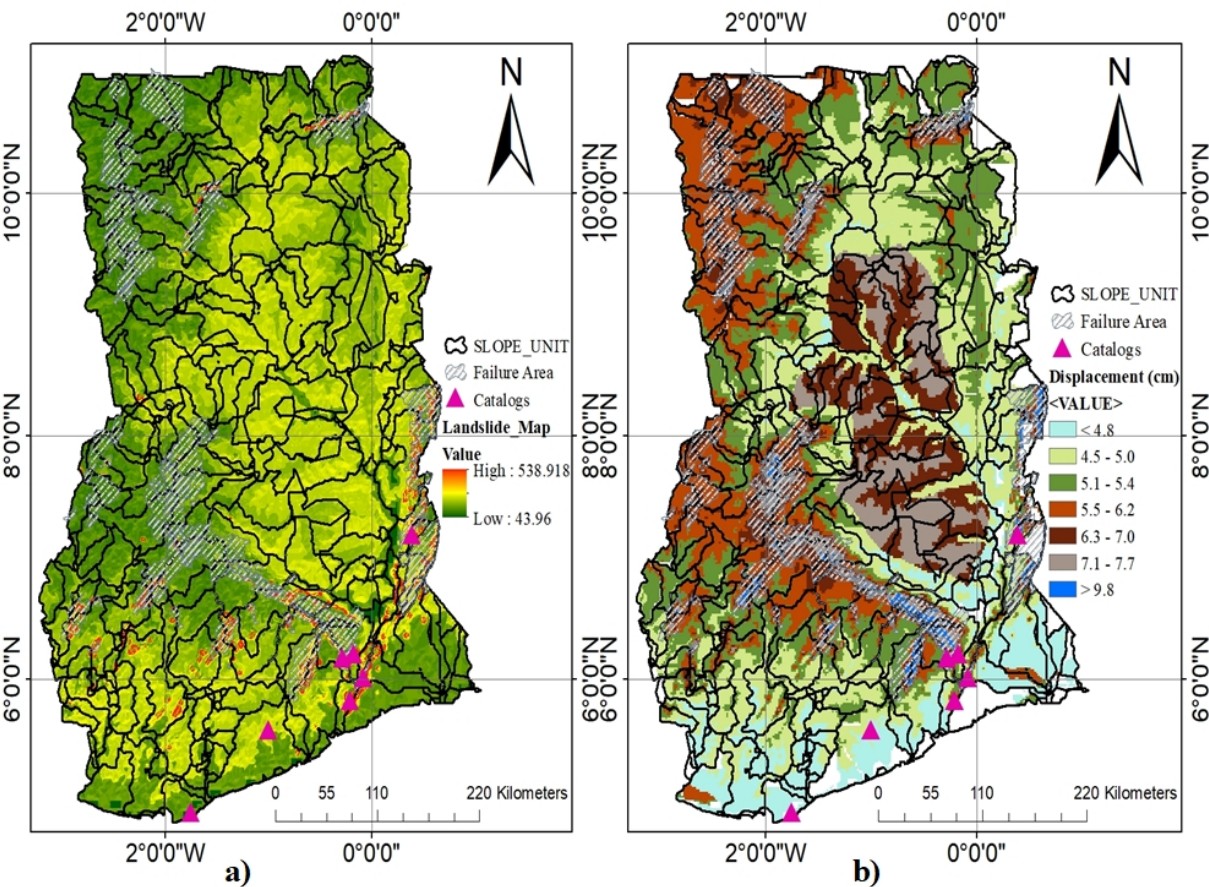

**Figure 10.** Displacement: a) Landslide susceptibility Map of Ghana b)Displacement map indicating failure areas and previous landslide catalogs

24 percent or below 11.3 percent of the total slope unit area.

$$FailureRate\,(F_r\%) = \frac{u}{a} \tag{22}$$

where $u$ is the area of slope unit liable to fail and $a$ is the total area of slope unit.

## 5   Sensitivity Assessment

### 5.1   Influence of slope unit

Unlike the Grid-cell method for displacement analysis, which is dependent on grid sizes of the DEM's pixel size and resolution, the analysis of the displacement threshold obtained through the study of the $P_r$ and rate of failure is the focus of the slope unit




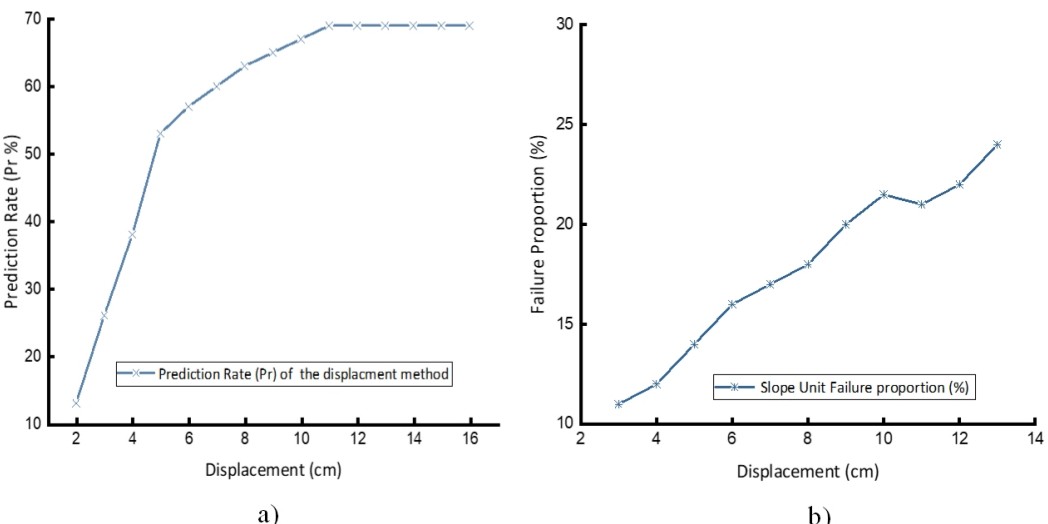

**Figure 11.** Displacement Analysis. a) Displacement Predicted rate (Pr) b) failure proportion

method for displacement analysis. Fulfillment of the analogy demands a comparison of the extracted slope unit in this research

Fig.6 (a) with slope unit extracted by the conventional method Fig.6 (b) and Hydraulic methods' slope unit Fig.6 (c).

The method used for this research (point segmentation by morphological slope unit method) produced 241 slope units of Ghana, whiles the conventional and hydraulic methods had 144 and 303 slope units, respectively. The traditional method produced slope units with larger unit areas and trailed the point segmentation by the morphological method by 40.2% regarding the number of slope units. The hydraulic method also produced a slope unit with 21% numbers more than the pour point

segmentation method and had a threshold displacement of around 9% in Fig.12, consistent with (Jibson & Keefer, 1993). The pour point segmentation method produced a slope unit with a threshold displacement of 9 cm (within the allowable) and a failure rate of 11.3 to 24, with regular and reasonable boundaries eliminating conjoint slope units.

## 5.2 Influence of strength parameters

Various researchers have enumerated the relevance of evaluating the influence of strength parameters on slopes' displacement.

Because assigning low strength parameters results in a higher displacement rate while giving high strength parameters also tend to underestimate the displacement, therefore influencing the results. (Dreyfus et al., 2013; Tsai et al., 2019) pointed that the lower bound strength (material having low strength parameters) tends to present a factor of safety below one ($F_S<1$) and thus needn't be considered in the analysis. In this regard, in addition to the medium bound strength parameters used for this research, in Table. 4, the parameters are further tilted for higher, medium, and low strength then used to evaluate the influence of

the material's strength on displacement Fig.13. It's observed that the strength of the prediction rate and failure proportion never really changed but instead followed a particular trend, Fig.13 confirming (Tsai et al., 2019). This indicates that, the material's strength does not affect the displacement $P_r$ but for the $k_y$(g). The study then evaluated the materials effect on acceleration by




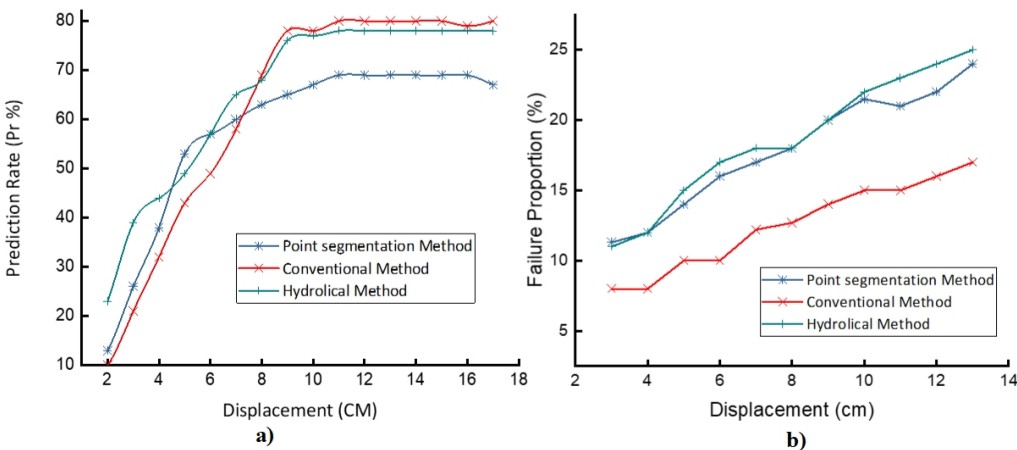

**Figure 12.** Sensitivity Assessment of different slope unit extraction methods. a) Displacement Predicted rate of different slope unit extraction methods. b) failure proportion of different slope unit extraction methods.

establishing a relationship between displacement and $k_y$(g). Ninety-nine percent correlations is obtained, echoing how strength properties affect the $F_s$ and the $k_y$ (g) but not the $P_r$ of displacement.

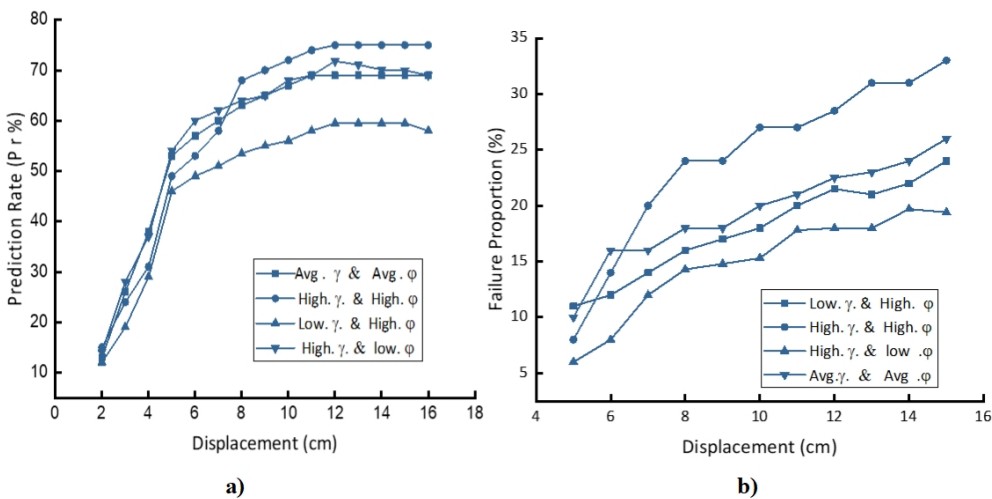

**Figure 13.** Sensitivity Assessment of different strength parameters a) Displacement Predicted rate of different strength parameters ($P_r$) b) failure proportion of different strength parameters





## 5.3 Influence of displacement model


Evaluating the effect of a displacement model used for analysis is vital. (Dreyfus et al., 2013) used the source cell method to evaluate the influence of different displacement models on the North-ridge earthquake landslide. (Tsai et al., 2019) also used the slope unit approach to analyze a displacement model's effect by comparing other displacement models $P_r$ to determine their impact. In this study, the influence of displacement is evaluated by comparing the $P_r$ of rigid displacement models by (Jibson et al., 2000; Tsai et al., 2019; Zhang et al., 2019) through slope unit- approach. It could be observed from Fig.14 that, all three model shows a similar range of displacement prediction. Although they have varying displacement thresholds, they fall within the acceptable range of 5 - 10 cm (Jibson & Keefer, 1993).


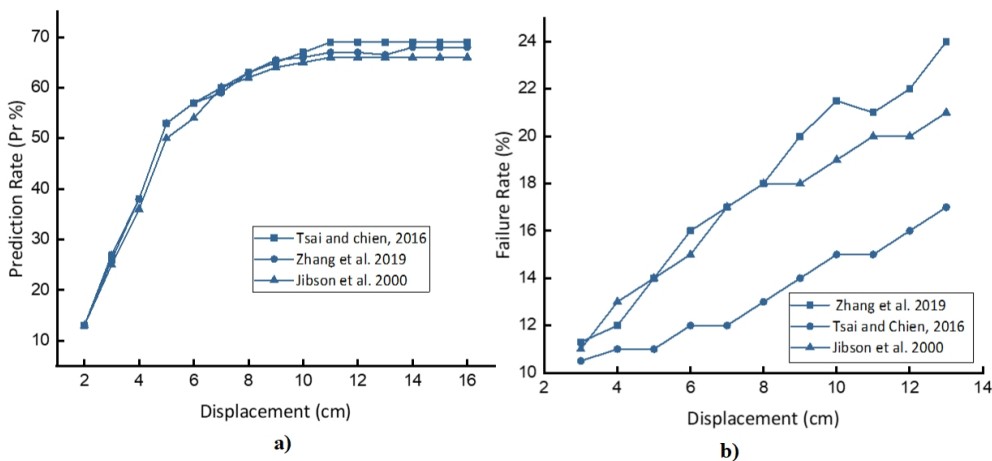

**Figure 14.** Sensitivity Assessment of different strength parameters a) Displacement Predicted rate of different strength parameters (Pr) b) failure proportion of different strength parameters

### 5.4 Influence of flexible and rigid mass

The rigid block assumes a shallow displacement failure, usually less than 3 m (Jibson et al., 2000; Tsai et al., 2019; Wang & Lin, 2010) and has been the current trend of consideration for most displacement analysis. Rigid block has been used for most displacement analysis regardless of the failure depth by considering the $T_s/T_m$ circumstance in Eq. (13). The displacement map gives diverse opinion for consideration because some of the areas are less than 0.1 ($T_s/T_m < 0.1$), and other places are also greater than 0.1 ($T_s/T_m > 0.1$). Therefore, to clear the doubt about rigid block and flexible mass displacement analysis consideration, the influence of both methods has been emphasized by analysing their Pr. The rigid block by (Zhang et al., 2019) in Eq.(16) and the flexible block proposed by (Rathje & Antonakos, 2011) in Eq.(18) are used in this study without any pre-assumptions. It could be seen from Fig.15 that neither the $P_r$ values nor the threshold displacement Values for both Rigid and Flexible displacement changed. This makes both the flexible and rigid block valid and influential for usage in displacement







analysis. Therefore it will always be appropriate to use the rigid block other than the flexible block, which seems a little complex in analysing.

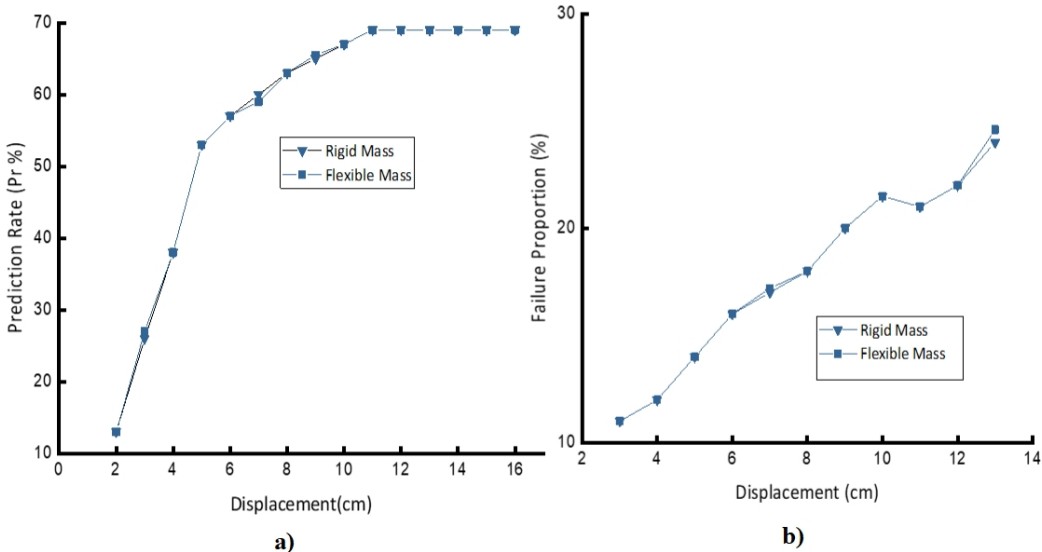

**Figure 15.** Sensitivity Assessment for rigid and flexible mass a) Displacement Predicted rate of flexible and rigid mass displacement b) failure proportion of the flexible and rigid mass displacement

## 345  6  Conclusions

Previous Regional landslide and displacement analysis were based on the grid-cell method, which sometimes ignores failure depth and slope geometry and relies solely on assumptions, these assumptions in most cases affects the accuracy of the results (Tsai & Chien, 2016; Tsai et al., 2019). The hydrological and conventional watershed slope unit methods also have problems with a sudden change in slope gradient (Wang et al., 2019), others like the curvature watershed method could also not extract
slope units beyond the water flow direction hence producing faulty slope units. Since this study's focus isn't to highlight the slope unit methods strength compared with the Grid-cell, other slope unit methods or the superiority of one displacement model to the other, a new slope unit extraction method is proposed in this research. The procedure for previous studies by (Tsai & Chien, 2016; Tsai et al., 2019) is followed to predict the slope's seismic displacement, neglecting cohesion in the analysis and considering both shallow and deep failure. The procedure is implemented on analyzing the possibility of slope displacement
in Ghana (West Africa). To have confidence in regional seismic displacement and hazard Maps, methodologies used to derive these maps need to be compared with and validated against field observations. The optimum threshold displacement that yields accurate prediction is 5 –10 cm, which accords to field observations and studies, making the procedure and methods used in this





research justified. The procedure is further perused by conducting a sensitivity analysis to evaluate the slope unit's influence, strength parameters, displacement models, and Rigid and Flexible Mass. The conclusions below are obtained:

1. slope unit highly influences landslides $P_r$. An increase in slope unit size causes a decrease in prediction rate and vice versa under the same condition.

2. Strength parameters do not influence $P_r$, but for the $F_s$ and $k_y$ (g).

3. Displacement models influence the $P_r$. Best displacement models might have slight changes in their output but produces the same optimum displacement threshold of 5-10 cm (Zhang et al., 2019). In this study, the displacement model
by (Zhang et al., 2019) performed better when compared with (Tsai et al., 2019). And both displacement models outperformed (Jibson et al., 2000) for the implementation of the study in Ghana. However, all their displacement models produced a good $P_r$ and acceptable displacement threshold of 5-10 cm, making all three displacement models viable for displacement analysis on regional scale. Further studies may be needed to confirm this suggestion.

4. The rigid and flexible block for shallow and deep failure ended up producing the same prediction $P_r$, which means the
Rigid block assumption by (Zhang et al., 2019) is valid compared with the flexible mass by (Rathje & Antonakos, 2011).

The procedure used for the landslide hazard assessment in this study is more feasible for the susceptibility and displacement mapping at both regional and national scales. It tends to predict more accurately for shallow than deep slope failure and presents the possibility of slope displacement in Ghana. The proposed method's accuracy cannot be undermined, as it proves its strength by presenting a prediction rate of 70.9% compared with the landslide inventory given the estimated displacements. The results
also demonstrated that a seismic magnitude above 7.9 could trigger slope displacement in Ghana. This work's essential data are DEM, Strength properties, seismic activity records, and geologic Data.

*Acknowledgements.* This study has received financial support from the National Natural Science Foundation of China (41977213, 41977235); Science Technology Department of Sichuan Province (2020YFH0017, 2021YFS0321); the Second Tibetan Plateau Scientific Expedition and Research Program (STEP) (2019QZKK0906). The financial supports are gratefully acknowledged.

**7  Appendix A**

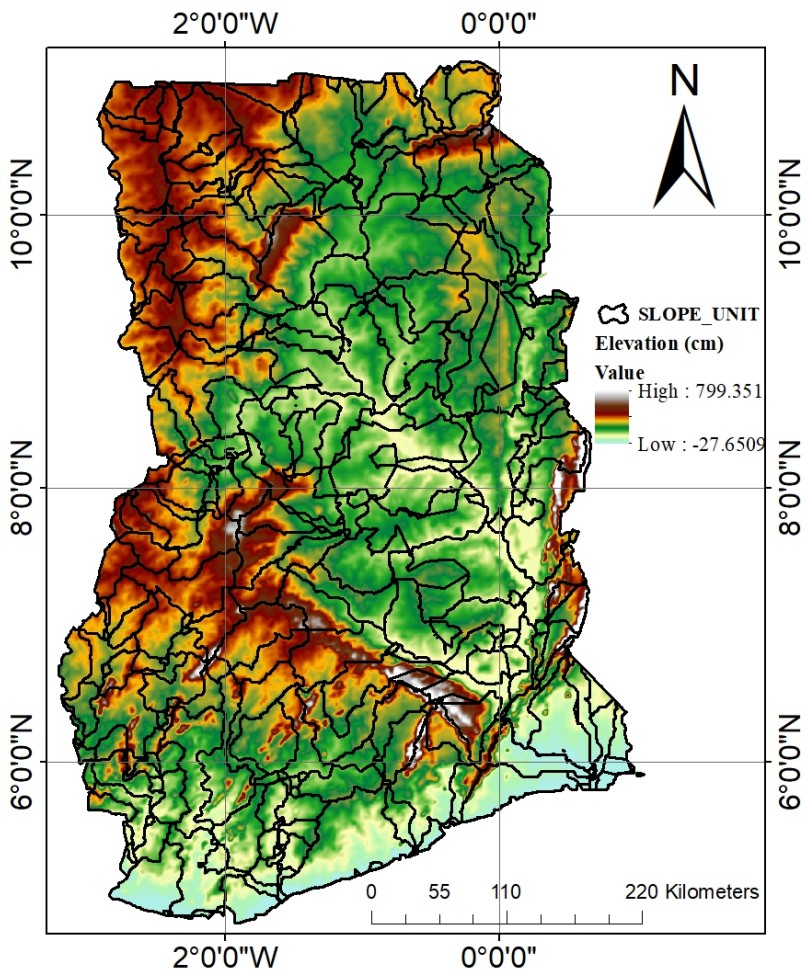

**Figure A1.** Re-sampled 30m*30m Resolution DEM Elevation map of Ghana

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
