# Peer review of "Hazard Assessment of Earthquake-Induced Landslides Based on a Mechanical Slope Unit Extraction Method, A Case in Ghana"

_Natural Hazards and Earth System Sciences, 2022_

## Author Comment (AC1)

**Entitled "**Hazard Assessment of Earthquake –Induced Landslides Based on a Mechanical Slope Unit Extraction Method**"**

15[th] April.2022

Dear Editor,

We sincerely appreciate the editor's/reviewers time and effort in evaluating our manuscript. We agree with and accept all the comments and suggestions from the Editor and Reviewers. We have carefully and thoroughly revised the manuscript according to the editors/reviewer's questions and comments. In the revised manuscript, changes are shown by using the track changes mode. The point-to-point responses to the comments are detailed as follows:

**Editor**

**Comment:** You as the contact author are requested to individually respond to all referee comments (RCs) by posting final author comments (ACs) on behalf of all co-authors no later than 04 Aug 2022 (final response phase).

**Response:** Thank you for your comment and suggestion.

We agree with your opinion. According to the Reviewers' comments and suggestions, the questions have been answered based on the thoroughly revised manuscript.

**Anonymous Reviewer (RC)#1**

The paper would benefit from a deep systematic revision of the presentation of results and proposed approach with respect to the claimed aim before consideration for publication in Natural Hazard and Earth System Science:

**Comment 1:** The Introduction chapter is unbalanced on the empirical literature relations available for the calculation of the coseismic displacement, and the different solutions for landslide stability analysis with respect to slope units' delineation approaches.

**Response:** Thank you very much for your constructive comment.

We agree with your comment. Following the reviewer's comments, the introduction has been overhauled to balance the literature and empirical relations available for calculating the coseismic displacement and the different solutions for landslide stability analysis, as written below,

**After revised: Introduction Page 1 (Line 18-24) and Page 1, (Line 1-35) -Manuscript**

Earthquakes are the most dangerous natural hazards, posing the most significant risk to life and property. Since the 1980*s*, earthquakes-induced landslides have caused many deaths and economic

losses. For example, the Chi-Chi earthquake-triggered about 9000 landslides (Tsai et al., 2000), and the Guatemala earthquake triggered 10,000 landslides

(Hamilton, 1997). Landslides mainly occur when acting forces exceed the strength of earth materials that composes a slope, and its evaluation provides general insight into future earthquake-induced landslides based on medium and long-term predictions of earthquake distribution to provide a possible mitigation measure to control its impact on life and properties (Bray et al., 2018; Salunkhe et al., 2017; Tsai & Chien, 2016; Wang & Lin, 2010; Zhang et al., 2019). Statistically-based methods based on historical landslide distribution were typically used in the early days for hazard zonation. However, the engineering approach (i.e., physically-based modeling) with the application of slope stability analysis models has recently been intensively studied and used to analyze landslides (Cencetti & Conversini, 2003; Tsai et al., 2019).

The statistically-based method could be bivariate (Chung & Fabbri, 2012; Chung & Fabbri, 2003; Dai & Lee, 2002; Wubalem, 2020), a multivariate method (Atkinson & Massari, 1998; Polykretis et al., 2019), Artificial Neural Network (*ANN*) method (Ortiz & Martínez-Graña, 2018; Tsangaratos & Benardos, 2014; Vakhshoori et al., 2019) or Machine Learning Techniques (*MLTs*) (Tien Bui et al., 2012; Youssef & Pourghasemi, 2021; Kavzoglu et al., 2014). Statistically-based methods assume that landslide controlling factors are conditionally independent of each other (e.g., Youssef et al., 2016). Hazard maps generated by some of these Statistically-based methods are obtained from a combination of maps generated using control points, whose predictive variables suffer from multicollinearity. The multivariate statistically-based methods are also suitable for large and complex areas. However, the method's robustness highly depends on the database used for the analysis. And only conditionally identical to those in the database can be predicted (Tien Bui et al., 2012; H. Y. Tsai et al., 2019). Engineering methods for earthquake-induced landslides and displacement analyses are done using the sliding block displacement method, which is a compromise in complexity between simple pseudo-static analysis and complex numerical simulation engineering methods (Ellen et al., 1998; Jibson & Keefer, 1993; Jibson et al., 2000; Saygili, 2008; Tsai et al., 2019; Wang et al., 2017; Shinoda & Miyata, 2017, Zhang et al., 2021). The sliding block method considers the landslide as a rigid non-deformable plastic body that slides on a plane of continuous dip and friction and only accurately predicts the displacement of an individual sliding body ( Jibson, 2011;   Tsai et al., 2019). The Newmark's sliding block method produces a stronger correlation between the estimated displacement and the mapping location of the earthquake-triggered landslide, making it a good engineering method suitable for predicting earthquake-induced landslides (Rathje et al., 1998; Tsai et al., 2019; Xie et al., 2003; Zhang et al., 2019). Newmark's sliding block method is the first for seismic-induced landslides' displacement analysis   (Jibson, 2011; Newmark, 1965). The Newmark's sliding block method is useful for rapidly predicting a seismic-induced landslide by first diving the study area into numerous grids, especially during regional displacement assessment (Tsai et al., 2019). These grids are assumed to be infinite, hence having definite depth (usually less than 3m ), and as well neglect slope geometry in their analysis,

therefore, modeled as rigid blocks (Ellen et al., 1998; Jibson et al., 2000; Xie et al., 2003; Zhang et al., 2019). Rathje & Antonakos, (2011) pointed out that ground motion parameters for displacement analysis on a regional scale can be altered to consider both shallow and deep failure due to their interaction with the sliding soil material. A framework for predicting earthquake-induced displacement is therefore proposed based on Zhang et al., (2019) to overcome the problems of defining slope displacements as an infinite sliding block with shallow depth, to consider it as a finite failure, especially in less cohesive soil materials whiles considering the effect of pore pressure and slope geometry in determining the safety factor $F_s$ to predict an unbiased $k_y(g)$ for the displacement analysis. This framework is based on regional hazard analyses; therefore, the study area must first be divided into sampling units of landslide hazard zones in which every landslide influence factor can be allocated. These landslide hazard zones are mapping units (Ba et al., 2018).

The most popular Mapping units for earthquake-induced landslide displacement analysis include the grid-cell, slope unit, etc. (Schlögel et al., 2018; Tsai et al., 2019; Yu & Chen, 2020). Grid cells are regular square cells with a given size for the unit mapping of landslides and are not closely related to geological environments (Guzzetti et al., 1995). As highlighted by Xie et al., (2003), a limitation of the grid-cell is its inability to represent natural slope topographic boundaries in their natural condition because it uses artificially marked cells of a block to represent the natural landscape event.

According to hydrological theory, a "slope unit" is considered a watershed defined by the ridge and valley lines and is used to divide spaces into smaller regions for easy analysis, making the method more related to the geological environment, hence the best for landslide and displacement analysis (Guzzetti et al., 1995; Wang et al., 2017). The slope unit is more applicable than the grid-cell method because landslides occur on slopes; therefore, the slope unit represents the topographic feature more thoroughly than the grid cell (Wang et al., 2017; Xie et al., 2003). Slope units are usually extracted from the digital elevation model (DEM) using geographic information system (GIS) software (Wang et al., 2019). The method for the extraction of the slope unit involves delineating a watershed from a DEM, then reversing the DEM to delineate another watershed. The two watersheds are merged to end the extraction of the slope unit (Mesut et al., 2011). This slope unit extraction method is termed the hydrological slope unit extraction method (Fig. 1a)(Mesut et al., 2011). Slope units extracted with the hydrological method are usually based on the surface hydrological process, making it impossible to identify variations in slope gradient beyond the hydrological flow direction, resulting in a sudden change in slope gradient. As such, slope units extracted using the hydrological methods suffer heterogeneity effects primarily associated with slope units extracted using high-resolution DEM (Guzzetti et al., 1995; Wang et al., 2019, 2020). The hydrological slope unit extraction method again produces irregular boundaries and conjoined slope conditions. This occurs because it barely distinguishes inclined and horizontal planes of deep valleys and high mountainous terrains (Wang et al., 2019). Tedious manual post-extraction corrections are needed to make the slope unit acceptable (Cheng & Zhou, 2018; Wang et al., 2020).

This research proposes a new slope unit extraction method using GIS software for the earthquake-induced displacement framework analysis. The method combines catchment points, hydrological slope unit extraction method, and segmentation to overcome the limitations of the hydrological slope unit extraction method. The application of the slope unit extraction method and displacement framework is validated in Ghana. The prediction result of the slope unit extraction method is compared with the hydrological method. The displacement framework is also compared with the displacement method by Jibson et al., (2000); Tsai et al., (2019). The paper also underlines the possibility of the proposed model for displacement analysis of shallow and deep slope failures considering the pore water pressure during the computation of the factor of safety $F_S$.

**Comment 2:** The authors stated from line 80 that the main aspect of the model used and proposed has three distinct features compared to the others and include: i) the SU delineation, ii) the consideration of pore water distribution and iii) the GIS computation of $F_s$ and the $ky$ (g) to avoid iterative errors: The first important aspect is the SU definition approach that should solve slope heterogeneity defects is not adequately presented and is not clear and easy to understand by the general audience. Secondarily, the role and the areal constraints of pore pressure (hydrostatic) lacks in the whole manuscript. Is it considered parametrically? Does the SF of Fig. 7 account for the $r_u$ pore pressure ratio reported in the methods section? Finally, the areal GIS-based quantification of $SF$ and $ky(g)$ appears to be a commonly used approach in the scientific community.

**Response:** Thank you very much for your constructive comment.
We agree with your comment. Following the reviewer's comments, we have overhauled the article, restructured it, and added some sentences and figures to enhance clarity on the heterogeneity defect in Fig.4, page 23, and Page 12 (Line 6-16) and 22 of the manuscript and presented it as stated below,

**After revised: (i) Slope unit heterogeneity [Fig.4, page 23 and Page 12, (Line 6-16) - Manuscript**

Slope angle is a critical factor for consideration in a landslide analysis ( Wang et al., 2019). Fig.4 (c) shows the terrain profile line $A'$-$A$ (convex area) with two flat slopes of $30^o$ and $10^o$ respectively, and a slope toe angle of $45^o$. Fig.4 (a) illustrates a slope unit extracted using the point segmentation method showing line A'-A area with three local slope unit regions. Fig.4 (b) is a slope unit map obtained using the hydrological method showing line A'-A (convex area) with two local slope unit regions and two angles. ArcGIS statistical tool is used to compute the slope angle of the slope unit region and profile in Fig.4 (a) and Fig.4 (b). Fig.11 (a) shows the terrain profile of the slope unit region determined using the hydrological method; the terrain shows just two slope angles, with the toe having a 45 angle and a second slope angle of 10 instead of 30 (as was demonstrated in Fig.4 (c)). This indicates that the hydrological method underestimates the slope angles compared to the point segmentation method. The point segmentation method divided the area into three slope units and three angles having the exact sizes and conforming homogenously to the terrain in Fig.4 (c).

[Figure]

**Fig. 4** Slope units of Ghana derived using different methods. a) Results from the point segmentation method b) Result from hydrological method c) Simplified terrain profile along *A''-A*. c) Simplified terrain profile along *A''-A* for hydrological method These slope units are overlain on elevation maps. The region enclosed by red lines indicates how the point segmentation method solves the irregular boundary defect and is further described in the sensitivity section. The region enclosed by white lines in a) and b) also indicates how the point segmentation methods solve the slope unit heterogeneity effect from the hydrological method and is further described in the sensitivity.

[Figure]

**Fig. 1** Profile of slope unit by different methods. a) *A'-A* Slope unit profile determined by the hydrological method and b) *A'-A* Slope unit determined by the point by the segmentation method

**After revised: (ii) The effect of pore water pressure has been adequately presented in this manuscript [Page 5, Line 7 -24]**

The aspect of the displacement method used has also been reviewed in the manuscript to include the effect of cohesion, however, the influence of pore water pressure remains unchanged and used in the manuscript during the determination of the safety factor,' $F_s$' Fig.6 (a). It's presented in the manuscript as inscribed below,

Most slope unit methods are typically adopted to estimate the Newmark-type displacement for regional seismic landslide hazard assessment. This procedure is limited to the assumption of infinite slope and rigid block movement, which may be different from the actual behavior of slope exposed to earthquake loading. A new slope unit-based approach is proposed in this study (point segmentation) to estimate earthquake-induced displacements for finite slopes. The method's main advantages are that it considers slope geometry and tends to automatically predict landslide hazards for both finite and infinite slopes. Using the method for finite slope displacement analysis means making considerations for the effect of the pressure associated with underground water that may affect it. Because When pore water pressure is neglected in the strength reduction of a finite slope, it may affect the excess pore water pressure that influences the yield acceleration $k_y$(g) and underestimate the displacement (Biondi et al., 2007; Sahoo & Shukla, 2019). The effect of pore water pressure at a depth within the soil or underground Hydrostatic consideration has also been adequately presented in this paper for the delineation of an $F_s$ which was used to compute for slope unit displacement and has been mentioned in the research as such Eq. (5) sums up the $F_s$ method used in this study for the delineation of Fig. 6(a).

$$F_s = \frac{c+(\gamma - m\gamma_w)d\cos\beta\tan\varphi}{\gamma d\sin\beta}[1-r_u] \qquad (5)$$

[Figure]

**Fig. 6** Automatic generated Maps a) Factor of Safety, $F_s$, b) Yield acceleration, $ky$,

**After revised: (iii) GIS-based quantification of Fs and ky(g), [ Page 4 (line 1-8)]**

*Delineation of the Fs and ky(g) using the GIS approach to eliminate iterative error seems common in some research studies on displacement analysis as stated by the reviewer. As such been changed and thus the distinct aspects of the study have been reviewed on page 4 of the Manuscript as follows,*

i) Extraction of slope unit that accurately predicts the watershed and morphology (aspect and gradients) of the study area. $3^D$ elevation watershed points of the study area were first derived before being used for the extraction of the slope unit. These points helped obtain accurate slope gradients, and ridge and valley lines were used in the extraction of the slope unit to eliminate heterogeneity defects associated, and segmentation is done to solve boundary defects associated with the hydrological method.

ii) The extracted slope unit considers slope geometry and tends to automatically predict landslide hazards for both finite and infinite slopes.

iii) An $F_s$ that considers the effect of pore-water pressure alongside the traditional Newmark's method that considers failure depth $d$.

**Comment 3:** The rigid and flexible block effect is considered important and treated analytically; however, the contribution of cohesion in earth shallow landslides is considered irrelevant, which are mainly governed by the effect of apparent and mechanical cohesion of unsaturated media. A critical comment about this topic would improve the manuscript. In addition, eq. (8) by Saygili & Rathje, 2009 is dependent on cohesion. Which values have been adopted?

**Response:** Thank you very much for your constructive comment.

We agree with your comment. Following the reviewer's comments, the rigid and flexible method has been modified to include cohesion as presented in the manuscript Page 7, (Lines 5-30)]. We initially ignored the effect of cohesion because, according to Biondi et al. (2007), consideration for the effect of pore water pressure during stability analysis of slope is usually done for cohesionless soil materials. As such, this study decided to base the displacement analysis on designing for the worse possible condition. Thus, neglecting the effect of cohesion. However, after further consideration based on the reviewer's comment, the study omitted the idea of neglecting the effect of cohesion in its analysis. And has included the effect of soil cohesion in the determination of the $F_s$, hence including the effect of cohesion in the displacement analysis as stated on page 7 of the manuscript

**Comment 4:** Are the landslides plotted in Figures seismically induced? In some figures are reported locations of "failure areas and catalog" that are not explained and/or not fully considered in the validation of results.

**Response:** Thank you very much for your question.

The failure areas and catalog are explained on Page 10 (Line 21-29) of the manuscript as written below,

**After revised: i) Failure areas [Page 10 (Line 21-29)-manuscript]**

The failure areas and catalogs are explained in the manuscript on page 10 (Line 21-29)of the manuscript.

Because Ghana (the Study area) lacks a strong landslide database that could be utilized to predict failed slopes within the slope unit, the failed regions in this study were predicted using the displacement map alongside the prediction accuracy method by Jibson & Keefer, (1993); Tsai et al., (2019). The method states that the prediction accuracy for slope displacement under seismic loading should depend on the relationship between the threshold and predicted displacements (noting an allowable threshold displacement between 5-10 cm depending on the residual slope). Using the displacement map in Fig.9 (b), areas within the slope units having predicted displacement above 10 cm were selected as the probably failed areas in this study    Jibson & Keefer, (1993). The predicted failed areas were consistent with the Landslide susceptibility map of Ghana in Fig.9(a) obtained using the Frequency Ratio $F_r$ method proposed by Buah et al., (2019).

[Figure]

**Fig. 9** Displacement: a) $T_s/T_m$ of Slope Unit in Ghana b) Landslide susceptibility Map of Ghana c) Displacement map indicating failure areas and previous landslide catalogs (epicenters)

**After revised: ii) Earthquake Epicenters/catalogs (Page 10 and Fig.8 – manuscript)**

On the other hand, the major earthquake catalogs /Epicenters considered in this research indicate whether the failure areas on the displacement map coincide with the Earthquake epicenters because the research assumes the displacements of slopes are seismically induced. However, soil properties play a role.

**Comment 5:** Regarding the Prediction rate, I'd spend more effort discussing the general concept and the meaning of the two curves for the validation of the results. With a few landslides' observations, I supposed the $P_r$ is overbalanced by a large number of negatives. I'll express the success rate as the ratio between the true positive rate and the false-positive rate. Would the success rate for true positives be more informative? What about the effect of strength parameters on the lone True Positives ($S_1$).

**Response:** Thank you very much for your suggestion.

The prediction rate ($P_r$) and Failure proportion (%) have been reviewed and adequately explained on pages Page 11 (line7-22), Table.4, and Fig.10 of the manuscript.

**After revised: i) Prediction Rate curves [Page 11 (line7-22), Table.4 and Fig.10 [manuscript]**

**The** $P_r$ and failure proportion curves have been adequately detailed in the manuscript as demanded by the reviewer as such,

The predicted and threshold displacement of the slope subjected to earthquake loading can be used to determine its failure. A threshold displacement of 5-10 cm is deemed reasonable (Jibson & Keefer, 1993; Tsai et al., (2019). In this situation, given a threshold displacement of 10 cm, all areas with slope units having predicted displacements >10cm are indicated as failed areas and used for the validation of the displacement through the prediction rate $P_r$ procedure (Wang & Lin, 2010). $P_r$ is a ratio of the slope

units with accurate predictions (true positives) to those with inaccurate predictions (false positives). The true positives encompass slope units containing failure scars and displacement greater than threshold displacement ($S_1$) and slope units without failure scars with displacement lower than the threshold displacement ($S_4$). The total number of slope units also includes the incorrect predictions (false positives), which also encompasses slope units without scars but displacement above the threshold ($S_2$), and those with scars and displacement less than the threshold ($S_3$) Table.4. The $P_r$ values for all assumed displacements are shown in Fig.10 (a). For the predicted displacement map in Fig.9 (b), the $P_r$ value is 35 %, with 85 accurate predictions out of 241 slope units at a threshold displacement of 5 cm. The Method also presents a 68% $P_r$ value with 164 accurate predictions out of 241 slope units at a threshold displacement of 10 cm. Per Fig.10 (a), the $P_r$ has a maximum value of 68.5 % with 165 correct predictions out of 241 slope units when the threshold displacement of slopes in Ghana is defined as 9 cm, which agrees with the suggestion by Jibson & Keefer, (1993)

$$P_r = \frac{S_1 + S_4}{S_1 + S_2 + S_3 + S_4} \tag{20}$$

**Table 4** Displacement prediction parameters $S_1$, $S_2$, $S_3$, and $S_4$

| No | Slope Unit Location identification | Symb | Accurate Prediction | Inaccurate Prediction | Values at 5 cm displacement | Val. at 10 cm displacement |
|----|-----------------------------------|------|--------------------|-----------------------|----------------------------|---------------------------|
| 1 | With scars and displacement above threshold | $S_1$ | $S_1$ | | 37 | 25 |
| 2 | Without Scar and displacement below threshold | $S_4$ | $S_4$ | | 48 | 139 |
| 3 | Without scars and displacement above threshold | $S_2$ | | $S_2$ | 114 | 77 |
| 4 | With scars and displacement below threshold | $S_3$ | | $S_3$ | 0 | 0 |

[Figure]

**Fig. 2** Displacement analysis for slopes in Ghana. a) Displacement $P_r$ plot showing the maximum $P_r$ value at 68.5% for the threshold displacement of 9cm, b) failure proportion plot showing the least portion of slope to fail in Ghana to be 9.3% and the

**After revised: ii) Failure Proportion (%) (page 11(line 24-27), Page 12(line 1-4)- Manuscript**

According to ( Tsai et al., 2019), a predicted slope failure does not necessarily indicate that the entire

slope within a slope unit will fail but rather a portion. Therefore, the proportion of predicted slope failure in Ghana is computed as, the ratio of failure within the slope unit, u, to the slope unit's entire area, an Eq. (21). In comparison to the $P_r$ value, the failure proportion is less affected by threshold displacement.   The result from Fig.10 (a) shows that the failure portion value of 9.3 to 17.8% means that even if a slope unit is considered to fail; only 9.3 to 17.8 percent of its area is at risk of failing. The predicted failure rate of slope in Ghana is likely to be between 9.3 to 17.8 percent. Where u is the area of slope unit liable to fail and a is the total area of slope unit.

$$Failure\ Rate\left(F_r\%\right)=\frac{u}{a} \tag{21}$$

**After revised: iii) $P_r$ Determination**

Thanks for your constructive comments supposing the $P_r$ is overbalanced by a large number of negatives. However, we have an opposing view, because defining the $P_r$ as the ratio between the true positives and the false positives will rather overbalance the results rather than the normal situation where the $P_r$ is a ratio of the true positives and the totality of the slope units. For example, using the 5cm displacement values in Table 4, the true positives sum up to 87 whiles the false positives sum up to 114 whiles the total slope units are 241. So, the normal procedure used for this research gives a $P_r$ of 36%, whiles when the ratio between the true and false positive values is used, a $P_r$ value of 76%, which seems to be too high for 5cm threshold displacement. Therefore, inappropriate to be used.

**Comment 6:** Concepts expressed in methods are often repeated in the results section. Technical language is often less precise (see Detailed comments). The introduction of the chapter "Seismic activity of Ghana" is mainly focused on African landslides, however, a detailed analysis of the available landslide catalog lacks. I would suggest changing the title or rephrasing the introduction to the chapter.

**Response:** Thank you very much for your constructive comment.
We agree with your comment. Following the reviewer's comments the introduction of the chapter "Seismic activity of Ghana" has been reviewed to focus on Ghana rather than Africa on Page 8(Line 9-30) of the manuscript.

**After revised: [Page 8 (Line 9-30)]**

"Ghana is far away from the major seismic zones of the World. However, the southern part of the country is seismically active and prone to earthquake disasters. Since the sixteenth century, places like Accra, Axim, Koforidua, and Ho have experienced seismic activities (Table 1). This seismicity is due to major faults in Ghana (Akwapim fault and Coastal boundary faults zones) connecting with West African continental tectonics (St. Paul and Romanche-transform fracture zone) offshore in the Gulf of Guinea to onshore (Blundell, 1976). As such, the tectonic activity of the Romanche transform fracture zone reactivates the seismicity on the Coastal boundary fault and causes earthquakes in places like Accra, Kasoa, Awutu-Senya, Weija-Gbawe, McCarthy Hills, and Adenta. Whiles the St. Paul fault

activities reactivate seismic activities from the Ivory Coast through Axim and intensify around the Akwapim fault zone through Koforidua and Ho. Ghana recorded its first earthquake in 1615 and the second one in 1636 in Axim, and all other subsequent ones are in Table.1(Amponsah et al., 2009). Seven severe earthquakes above 5.0 magnitudes have since struck the country in 1636, 1788, 1862,18791906,1907, and 1939. Seismic activities on the St. Paul's fault zone halted in 1879, making the Axim and Akuapim fault zone area free from reactivation of earthquakes. However, movements along the Romanche transform fracture zone fault are still in progress, making Accra and its environs vulnerable to seismic activities and tremors (Kutu, 2013). The western part of Accra (weija), on the junction of the coastal boundary and the Akuapim fault, has experienced most of the earthquakes in Ghana, making it the epicenter of earthquakes (Bates, 1962).

Ghana's landscape has low and high lands, with a rainfall pattern for a minimum of five months per annul. Some periodic earthquakes record has forced *GhIG* to predict the likelihood of a massive slope landslide in Ghana. The maximum intensity of the earthquake in Ghana is *IX* on the *MSK* Scale, recorded in 1862 (Ambraseys & Adams, 1991). Ghana's highest Peak Ground Acceleration (*PGA*) recorded was at the Accra-Tema seismic zone, estimated at 0.2g and minimized to 0.05g 140 km away from Accra. Ghana's Peak Ground Velocity (*PGV*) ranges from 9.2 to 37.1 cms-1, and the standard PGA ranges from 0.14 to 0.2g (Amponsah et al., 2009). In Ghana, areas with low *PGV* usually display high *PGA* (Amponsah et al., 2009)".

**Comment 7:** The manuscript is not adequately cared for in style and formatting. Figure and not sequentially cited in the text and often placed in the wrong sections. Many of them are not useful or simplistic. Reference to figures and table are not consecutive, captions not informative, and not self-standing. Please consider a detailed review according to Journal standards.

**Response:** Thank you very much for your constructive comment.
We agree with your comment. Following the reviewer's comments, The Manuscript is now adequately formatted by rearranging and restructuring the sentences and figures to suit the Journal's standard. References to figures and table are very consecutive. Captions are informative and self-standing.

**Comment 8:** Line 20: Please change the typos sentence that is supposed to be "in order to".

**Response:** Thank you very much for your constructive comment.
We agree with your comment. Following the reviewer's comments, the sentence has been reviewed on Page 1 (Line 24) of the manuscript. The new sentence reads "earthquake distribution to provide a possible mitigation measure"

**Comment 9:** Line 36: Newmark instead of Newark.

**Response:** Thank you very much for your constructive comment.

We agree with your comment. Following the reviewer's comments, the sentence on Page 2 (Line 18) "Newmark" has been correctly rewritten in the manuscript as "Newmark".

**Comment 10:** Line 60: Please deeply explain the limitation in reflecting morphological features.

**Response:** Thank you very much for your question.

The limitation of the morphological features is explained on page 3 of the manuscript. Morphological features are topographic landforms on the Earth's surface that require high-resolution elevation data to model them.

A slope unit is defined as the geomorphology of an area segmented into smaller mapping units by the ridge and valley lines or right and left-hand sides of a watershed sub-basin. An extracted slope unit must accurately represent the geomorphology of the study areas. A slope unit is as good as the method used in its extraction. If the method used is not suitable, the slope unit extracted wouldn't accurately represent the topographic features. As such, it doesn't reflect the exact geomorphology.

For example, slope units extracted using the hydrological slope unit extraction methods have sudden gradient changes along the flow direction. This affects the boundary of the extracted slope unit and the occurrence of slope unit heterogeneity. The slope unit extracted does not genuinely represent the study area's topography, therefore "not accurately reflecting the study area's geomorphology."

**Comment 11:** Do the authors referred to river thalweg with the term "crevasses"?

**Response:** Thank you very much for your correction. The sentence has been corrected on Page 4 (Line 23) of the manuscript and now reads,

**After revised: [Page 4 (Line 23)] - Manuscript**

"The mountain and river thalweg watersheds obtained are merged and segmented to delineate the boundaries before the morphological ridge and valley lines are determined to end the slope unit extraction".

**Comment 12:** Line 116: It is not tensile. It corresponds to the shear stress component parallel to the failure surface

**Response:** Thank you very much for your correction. The Correction has been made on Page 5 (Line 12) of the manuscript and the sentence reads.

**After revised: [Page 5 (Line 12)] - Manuscript**

"$\tau$" is the shear stress component parallel to the failure surface".

**Comment 13:** Line 121: "m" should be formatted in italics.

**Response:** Thank you very much for your correction *"m"* has been italic as indicated by the reviewer on Page 5 (Line 18) of the manuscript.

**Comment 14:** References are cited twice.

**Response:** Thank you very much for your correction, one of the references has been removed. (Page 3 (Line 3)of the manuscript) . (Dunn et al., 2011; Terzaghi et al., 1996)

**Comment 15:** Line 143: Please rephrase the definition of $k_y$(g).

**Response:** Thank you very much for your correction, we agree with your comment. Following the reviewer's comments, $k_y$(g) has been redefined on Page 6 (Line 6-8) of the manuscript. The newly written $k_y$(g) is as follows,

**After revised: (Page 6, Line 6-8)- Manuscript**

**Yield acceleration, $k_y$(g)**: Permanent displacing of the slope is mainly influenced by its yield acceleration properties, including groundwater level, geometry, material strength, and seismicity. In the sliding block procedure, $k_y$(g) is defined as the point where sliding failure or permanent displacement of slope initiates (Kan & Taiebat, 2005) Fig. 6(b) using Eq. (9).

**Comment 16:** Line 190: The list of coordinates can be omitted in the text since they are included in the figures.

**Response:** Thank you very much for your correction, we agree with your comment. Following the reviewer's comments, the coordinates of Ghana on Page 8 of the manuscript under the sub-heading Case Study – Ghana (West Africa) have been removed from the sentence.

**Comment 17:** Line 192, 194: Please use the International System of Units.

**Response:** Thank you very much for your correction, we agree with your comment. Following the reviewer's comments, International system units have been given to the rainfall in Ghana by changing the unit from centimeters (cm) to millimeters (mm) on Page 8 (Line 5) of the manuscript. And the sentence is written as,

**After revised: (Page 8, Line 5)- Manuscript**

"Ghana is tropical and the southern part and eastern part of the Volta Lake experiences the highest rainfall range between 1,400 to 2000 millimeters per year, from April to mid-November".

**Comment 18:** Line 148: Reference of DCF is missing. The factor is reported sometimes in subscript,

please uniform it.

**Response:** Thank you very much for your constructive comment, Following the reviewer's comments, The reference to the DCF has been added on Page 6 (Line 11) of the manuscript as "( Tsai et al., 2019)". The factor which was in subscript has been uniformed.

**After revised: (Page 6, Line 10-14) - Manuscript**

$$DCF = \begin{cases} \exp^{(0.4+0.343\tan\varphi)} \times \dfrac{D}{H} - 1.5 \times \left(\dfrac{D}{H}\right) & (\beta - \alpha \geq 5) \\ 0 & (\beta - \alpha < 5) \end{cases} \tag{10}$$

**Comment 19:** Line 205: The Romanche Transform fault and earthquake epicenters should be located on Map.

**Response:** Thank you very much for your constructive comment, Following the reviewer's comments, the Romanche faults and earthquake epicenters have been located on maps (Fig.7). (Page 26)

**Comment 20:** Line 217: Reference to figure missing.

**Response:** Thank you very much for your comment, there is no figure on line 217.

**Comment 21:** Unit weight cannot be considered a strength parameter.

**Response:** Thank you very much for your constructive comment, Following the reviewer's comments, the unit weight of soil ($\gamma$) has been replaced by cohesion as a strength parameter on Page 9 (Line 17) of the manuscript. And the sentence now reads,

**After revised: (Page 9, Line 17) – Manuscript**
"The required strength parameters used include the $c$ and $\varphi$"

**Comment 22:** Annum

**Response:** Thank you for pointing this out, Following the reviewer's comments, Annum has been correctly spelt in Table 1, Page 36 of the manuscript. And it's written as,
"Akoto and Anum, 1992, AKO".

**Comment 23:** Line 323: "Ninety-nine percent correlation is obtained". I missed the presentation

**Response:** Thank you for your question. The full statement is written on Line 323 of the manuscript (preprint version)

"The study evaluated the strengths of materials' effect on acceleration by establishing a relationship between displacement and $k_y$(g). Ninety-nine percent correlation is obtained, indicating how strength properties affect the Fs and the $k_y$(g) but not the $P_r$ of displacement". However, after critical considerations based on the review comments, the statement has been removed from the revised manuscript.

**Comment 24:** Table 2: In the percentage column 1/3 has been changed to 3/Jan by the corrector.

**Response:** Thank you for your constructive comments, the previous Table 2 has been removed from the revised manuscript.

**Comment 25:** Most of the figures are not informative (Esp. Fig. 4 or Fig. 5) and can be deleted.

**Response:** Thank you very much for your constructive comment.
We agree with your comment. Following the reviewer's comments, Fig.4 has been removed from the revised manuscript, and Fig.5 (now Fig.7) has been improved accordingly to contain the earthquake epicenters in Ghana (Catalogs) and the romance fault. (Page 26- manuscript)

**Comment 26:** Fig.7a, the red and green color bar looks counterintuitive to express the SF of slopes.

**Response:** Thank you very much for your constructive comment.
We agree with your comment. Following the reviewer's comments, Previously Fig.7a (now Fig.6a) "$F_s$ map" has been improved to vividly express the safety factor ($F_s$) of the slopes. (Page 25- manuscript).

**After revised: Page 25- Manuscript**

[Figure]

**Fig. 6** Automatic generated Maps a) Factor of Safety, $F_s$, b) Yield acceleration, $k_y$(g),

**Comment 27:** Fig. 14 caption is wrong, looks like a repetition of fig 13

**Response:** Thank you very much for your constructive comment.

We agree with your comment. Following the reviewer's comments, Caption for **Fig.3** has been reviewed to read "Sensitivity Assessment of point segmentation slope unit extraction method using Zhang et al., (2019) a) Displacement Predicted rate of different strength parameters ($P_r$) b) failure proportion of different strength parameters." (Page 32 - manuscript)

Caption for *Fig*. 4 has also been reviewed to read" Comparing different displacement methods a) Displacement Predicted rate of different displacement methods b) Failure proportion of different displacement methods" (Page 33- manuscript).

**After revised: Page 33- Manuscript**

[Figure]

**Fig. 5** Sensitivity Assessment of point segmentation slope unit extraction method using Zhang et al., (2019) a) Displacement Predicted rate of different strength parameters ($P_r$) b) failure proportion of different strength parameters.

[Figure]

**Fig. 6** Comparing different displacement methods a) Displacement Predicted rate of different displacement methods b) Failure proportion of different displacement methods

**Comment 28:** Table 2 can be merged with Table 3 since it is not informative. The real percentage distribution of the two major complexes is not indicative. I'd report it for the lithological unit in Table 3

**Response:** Thank you very much for your constructive comment.

We agree with your comment. Following the reviewer's comments, Table 2 and Table 3 have been merged with Table 4. The merged table is now labeled Table 2. The new "Table 2. Detailed geologic Unit, strength parameters and their respective locations in Ghana" is located on Page 37 of the manuscript.

**After revised: Page 37 -Manuscript**

*Table 2* Detailed geologic Unit, strength parameters, and their respective locations in Ghana

| Rock Unit | Abb. | Parameters | Fri. Angl. $\varphi(^o)$ medium | High $\varphi(^o)$ | Low $\varphi(^o)$ | Unit wt. $(\gamma)$ kN/m$^3$ medium | C. MPa | Location | Source |
|---|---|---|---|---|---|---|---|---|---|
| Precambrian | PC | Volcanic plutonic alkaline | 50 | 55 | 40 | 26 | 25 | Kumasi, sefwi, brong, central, wassa, some part of the north | Bohne & Frickie 1970 |
| Ordovician - Cambrian | OCM | Sedimentary, Mudstone and Siltstone | 45 | 50 | 40 | 25 | 32 | Volta Basin | Hoek & Bray 1989 |
| Water | H20 | | | | | | | Volta lake, Kumasi, tarkwa | |
| Holocene | QE | Granatoids of igneous rocks | 45 | 50 | 40 | 28 | 21.2 | Accra | Hoek & Bray 1989 |
| Tertiary | T | Quazite, shales & Granites of Igneous Rocks | 45 | 50 | 40 | 28 | 70 | Enchi and its environs. | Goodman 1980 |
| Quaternary & Tertiary | QT | Sand-stones | 30 | 34 | 27 | 24 | 27 | Kwahu & volta | Duncan and Norman |
| Carboniferous & Devonian | CD | Sedimentary Mud and Silt stone | 34 | 39 | 29 | 18 | 21 | Takoradi, secondi, axim | Goodman 1980 |

**Comment 29:** Fig. 5 can be merged with fig. 6.

**Response:** Thank you very much for your constructive comment.

We agree with your comment. Following the reviewer's comments, Fig.5 cannot be merged with Fig.6, because Fig.5 is a locational map while Fig.6 is a Slope Unit Map. Fig.5 (Now Fig.7) has been improved to show the geological details, major faults, and major earthquake epicenters in the study area (Ghana). The new Fig. 7 is located on Page 26 of the manuscript.

**After revised: Page 26 -Manuscript**

[Figure]

**Fig. 7** Major Lithostratigraphic and Lithotectonic Complexes of Ghana

**Comment 30:** Fig. 6 can be improved including the relief map reported in Appendix Fig. A1.

**Response:** Thank you very much for your constructive comment.

We agree with your comment. Following the reviewer's comments, Fig. 6 (now Fig.4) has been revised to include the elevation map in Appendix Fig.A1 and improved as such. An additional feature showing the slope unit profile (Fig.4c) is added and used to explain the slope unit heterogeneity effect. Located on Page 26 of the manuscript.

**After revised: Page 23-Manuscript**

[Figure]

**a)**          **b)**

**c)**          **d)**

*Fig. 4* Slope units of Ghana derived using different methods. a) Results from the point segmentation method b) Result from hydrological method c) Simplified terrain profile along *A"-A*. c) Simplified terrain profile along *A"-A* for hydrological method These slope units are overlain on elevation maps. The region enclosed by red lines indicates how the point segmentation method solves the irregular boundary defect and is further described in the sensitivity section. The region enclosed by white lines in a) and b) also indicates how the point segmentation methods solve the slope unit heterogeneity effect from the hydrological method and is further described in the sensitivity.

**Comment 31:** It is difficult to appreciate the difference between the three adopted models.

**Response:** Thank you very much for your constructive comment.

We agree with your comment. Following the reviewer's comments, the displacement of Ghana generated using different methods has been reviewed to show clearly the difference in the adopted models on Page 27 of the manuscript ("Then Fig.9", now Fig.8). The review is in accordance with the

new $F_s$ and $k_y$(g) map.

[Figure]

***Fig. 8*** Displacement maps: a) Tsai & Chien (2016); b) Jibson et al (2000); c) Zhang et a., (2019)

From the figure above it could be seen that the displacement method used for this research (Zhang et al., 2019) produced a displacement map having predicted failure areas in equivalence to the map prepared using the displacement method by Jibson et al., (2000); H. Y. Tsai et al., (2019). This displacement map looks alike with an error margin of about 1% percent. Confirming the accuracy of the prediction method compared with others.

**Comment 32:** Fig.12 Change "Hydrolical"

**Response:** Thank you very much for your constructive comment.

We agree with your comment. Following the reviewer's comments, Caption Hydrolical on *Fig*.12 has been changed to hydrological. And can be found on page 31 of the manuscript.

[Figure]

*Fig. 7 Comparative Assessment of different slope unit extraction methods, which looks different because different slope unit methods produced different flow accumulations a) Displacement Predicted rate of different slope unit extraction methods. b) Failure proportion of different slope unit extraction methods.*

**Comment 33:** Buah et al. 2019 is not listed in the reference

**Response:** Thank you very much for your constructive comment.

We agree with your comment. Following the reviewer's comments, Buah et al 2019 have been included in the listed reference.

**After revised:    References**

"Buah, P. A., Yingbin, P. Z., Bakah, D. A. Y., Ahiabu, M. K., & Zhibin, L. (2019). Earthquake-Induced Landslide Susceptibility Analysis : The Effect of DEM Resolution. International Conference on Mechatronics, Remote Sensing, Information Systems, and Industrial Information Technologies, ICMRSISIIT 2019. https://doi.org/10.1109/ICMRSISIIT46373.2020.9405915"

---

## Author Comment (AC3)

**Response letter to MS No: nhess-2022-43- Decision**

**Entitled "**Hazard Assessment of Earthquake –Induced Landslides Based on a Mechanical Slope Unit Extraction Method**"**

15[th] April.2022

Dear Editor,

We sincerely appreciate the editor's/reviewers time and effort in evaluating our manuscript. We agree with and accept all the comments and suggestions from the Editor and Reviewers. We have carefully and thoroughly revised the manuscript according to the editors/reviewer's questions and comments. In the revised manuscript, changes are shown by using the track changes mode. The point-to-point responses to the comments are detailed as follows:

**Editor**

**Comment:** You as the contact author are requested to individually respond to all referee comments (RCs) by posting final author comments (ACs) on behalf of all co-authors no later than 04 Aug 2022 (final response phase).

**Response:** Thank you for your comment and suggestion.

We agree with your opinion. According to the Reviewers' comments and suggestions, the questions have been answered based on the thoroughly revised manuscript.

**Anonymous Reviewer (RC)#2**

The manuscript has a reasonably good motivation. However, there are some fairly large points that I believe need further consideration. Below, I outline two main points to address before giving some detailed line-by-line comments:

**Comment 1:** Unfortunately, the manuscript was not written well and this effects my revision as a whole because I had a hard time following the flow and messages the authors are trying to convey. There are some terms (e.g., conventional, hydrological, and mechanical methods) that are not really well explained but continuously being used throughout the manuscript. I am sure the authors are clear about what they are referring to but unfortunately the same is not valid for the readership. Also, the structure of the manuscript is a bit "out of ordinary". There is no flow like introduction, study area, data, method, results, discussion, and conclusions. This also makes the text difficult to follow.

**Response:** Thank you very much for your question. This question has three parts and has been answered as such below

i) Message and flow of the manuscript

Landslide hazards can be predicted before it occurs or can be analyzed after it has already happened. This

analysis or prediction can be determined on a minor or regional scale. To determine how susceptible an area is to landslides on a regional scale, the site must first be divided into smaller units termed "mapping units." These mapping units could be based on grids (Grid cell) or slopes (slope unit). The research used the slope unit method for the susceptibility analysis because landslides occur on slopes (Xie et al., 2003). the slope unit approach involves extracting slope units which could be done using many methods. The simplest among the methods many researchers use is the conventional and the hydrological process methods. These two methods also have drawbacks (Wang et al., 2019). To rectify these drawbacks, a new slope unit extraction method is proposed (mechanical method) termed point by segmentation slope unit extraction method.

After obtaining the slope units, there is a need to predict or analyze the possibility of slope displacement. The most appropriate displacement method is Newmark's sliding block displacement. The Newmark's sliding block is a rigid block method appropriate for infinite slope units with definite depth (below 3m). however, Rathje & Antonakos (2011) pointed out that ground motion parameters for displacement analysis or prediction on a regional scale can be altered to consider both shallow and deep failure due to their interaction with the sliding soil material. A framework for predicting earthquake-induced displacement is therefore proposed based on Zhang et al. (2019) to overcome the problems of defining slope displacements as infinite sliding blocks with shallow depth, to consider it as a finite failure, especially in less cohesive soil materials whiles considering the effect of pore pressure and slope geometry in determining the safety factor Fs.

The prosed slope unit and displacement methods are tested in Ghana (West Africa) to determine how susceptible Ghana is to landslides. Although they have no strong landslide database, they have a record for minor landslides and earthquakes. A predictive landslide inventory based on Jibson & Keefer (1993); Tsai et al. (2019) is used. The method states that the prediction accuracy for slope displacement under seismic loading should depend on the relationship between the threshold and predicted displacements (noting an allowable threshold displacement between 5-10 cm depending on the residual slope). Using the displacement map in Fig.9 (b), areas within the slope units having predicted displacement above 10 cm was selected as the probably failed areas in this study and used to validate the research in Ghana.

     ii) Conventional, Hydrological, and Mechanical methods (Explanation)

Slope unit and all other terms have been explained in the introduction section of the reviewed manuscript on page 3, lines 4-16 as,

A slope unit is the left or right band of a sub watershed usually extracted from the digital elevation model (DEM) using geographic information system (*GIS*) software (Wang et al., 2019). The method for extracting the slope unit involves delineating a watershed from a DEM, then reversing the DEM to delineate another watershed. The two watersheds are merged to end the extraction of the slope unit (Cao et al., 2011). This slope unit extraction method is termed the hydrological slope unit extraction method (*Fig. 1a*) (Cao et al., 2011). Slope units extracted with the hydrological method are usually based on the

surface hydrological process. This makes it impossible to identify variations in slope gradients beyond the hydrological flow direction, resulting in a sudden change in slope gradient. As such, slope units extracted using the hydrological methods suffer heterogeneity effects primarily associated with slope units extracted using high-resolution *DEM* (Guzzetti et al., 1995; Wang et al., 2019, 2020). The hydrological slope unit extraction method produces irregular boundaries and conjoined slope conditions. This occurs because it barely distinguishes inclined and horizontal planes of deep valleys and high mountainous terrains (Wang et al., 2019). Tedious manual post-extraction corrections are needed to make the slope unit acceptable (Cheng & Zhou, 2018; Wang et al., 2020).

*After revised*: (*Page 3, lines 4-1) -Manuscript*

[Figure]

*Fig. 1* Slope Unit and slope types a)*3D* diagram of slope unit definition from the hydrological method (reverse DEM=DEM rotated by 180along the horizontal plane *A-A*). The number 1 represents a sub-watershed obtained from the DEM data, and the numbers 2 and 3 represent the watersheds obtained by reverse DEM calculations, b) *2D* view of circular slope failure model under static and failure mode, where *β* is the slope angle with ground and α is the slope angle of failure. c) *2D* View of Slope in Plane Failure Mode, where w is the weight of soil mass, *L* is the length, and t is the depth of the slope.

The term conventional has been removed from the revised manuscript.

However, the conventional method is also the same as the hydrological method and involves manipulations in *GIS*, including flow direction, flow accumulation, calculations, and catchment delineation. The *DEM* is reversed, and the procedure is repeated. the two catchments are merged to end the slope unit extraction method   (Cao et al., 2011). The term mechanical slope unit extraction has also been replaced with "a new slope unit extraction method" in the revised version of the manuscript

     iii) Structure

The structure of the manuscript might not be as expected by the reviewer. However, the topic under consideration and the procedure involved justify the process used (Tsai et al., 2019). The structure is as follows:

Introduction, framework for the proposed methods, case study (study area), method implementation and data, sensitivity (results and validation), and conclusions.

**Comment 2:** I am quite surprised to see that there is no landslide inventory or a specific earthquake the authors examined but still the manuscript is evolving around earthquake-induced landslides. I still do not understand how this could be possible. I might have missed something because of the reasons I mentioned above. However, taking this possibility aside, I do not understand the rationale of the manuscript. If there are no landslides triggered by earthquakes, the whole premise of the manuscript is just hanging in the air. I was planning to provide comments in detail for the entire manuscript but after realizing these issues mentioned above, I cut it short because, with all due respect to the authors' labor on the manuscript, I have to say those small revisions would not be adequate to make it publishable. Still, below I've included some of my line-by-line suggestions.

**Response:** Thank you very much for your constructive comments.

Ghana has had a series of earthquakes since 1615. The highest intensity of the Earthquake in Ghana is *IX*, with a maximum magnitude of 6.8mw on the *MSK* Scale, recorded in 1862 (Ambraseys & Adams, 1991). Ghana is a third-world country and does not have a substantial earthquake-induced landslide database. However, Ghana experienced rockfalls in the Aburi mountains in 2019 (myjoyonline. com).

" Landslide scare hit residents of Weija-Kasoa ridge in Ghana after a torrential rain which caused some minor landslide in the area (myjoyonline.com 24th May 2022). Because "Weija-Kasoa ridge" sits on a very active fault (Romanche-transform fracture zone), which has been seismically active since 1788 and reactivated on 25th June 2020 with an earth tremor having magnitude 4.2. This indicates that Ghana is prone to earthquake-induced landslides in the immediate or near future. So, in this study, since Ghana doesn't have a solid earthquake-induced landslide database, the authors used the displacement method to predict the possibility of earthquake-induced landslides in Ghana. The probably failed inventories from the displacement maps are used to validate the proposed slope unit and displacement method in Ghana.

**Comment 3:** Lines 17-18: "Landslides occur on slopes and have been the rationale behind making earthquake-induced landslides and seismic engineering a scientific and national demand". It is not clear what you mean here.

**Response:** Thank you very much for your question. The sentence has been reviewed in the revised version of the manuscript (Page 1 Lines 21-23) to read as "Landslides mainly occur when acting forces exceed the strength of earth materials that composes a slope, and its evaluation provides general insight

into future earthquake-induced landslides based on medium and long-term predictions of earthquake distribution to provide a possible mitigation measure to control its impact on life and properties".

**Comment 4:** Line 18: "its evaluation provides general estimates of future earthquake-induced landslides" Do you mean the evolution of earthquake-induced landslides? What do you mean? Or are you referring to the evolution of techniques that we use to assess landslide hazards or something? This line is not clear. Please revise and express it in a clearer way

**Response:** Thank you very much for your question. We mean to say that, evaluating already occurred landside is a means of acquiring insight into probable future occurrences of landslide (at the same or different location) to provide a means of mitigating the occurrence or curb its impact on life and properties.

**Comment 5:** Line 21: You can still examine historical landslide catalogs and address some research questions even today, right? What do you mean? Do you think it is not an old fashion approach?

**Response:** Thank you very much for your question. We mean to say that, evaluating already occurred landside is a means of acquiring insight into probable future occurrences of landslide (at the same or different location).to strategize means of mitigating future occurrences if possible or curb their impact on life and properties.

**Comment 6:** Lines 22-23: "current scientific and engineering stability analysis models" What are you referring to? Could you please be more specific?

**Response:** Thank you very much for your question.
In the early days, there wasn't the availability of physical-based modeling for landslide analyses. Therefore statistically-based methods based on historical landslide distribution were usually applied to hazard zonation. However, in recent times engineering approaches (i.e., physically-based modeling) with the application of slope stability analysis models have been intensively studied and used to analyze landslides (e.g., Jibson and Keefer, 1993).

**After revised: (Page 2, Line 1-3)- Manuscript,**

Statistically-based methods based on historical landslide distribution were typically used in the early days for hazard zonation. However, the engineering approach (i.e., physically-based modeling) with the application of slope stability analysis models has recently been intensively studied and used to analyze landslides.

**Comment 7:** Line 24: "The statistical method" Are you referring to a statistically-based method developed to assess landslide susceptibility? What sort of method are you referring to? Please be more

specific. Also, from lines 24 to 28, you do not say "such as" each and every time. You can just say, for instance: multivariate statistics (i.e., Logistic Regression, *LR*; Atkinson & Massari, 1998) Btw, I also noticed that you do not refer to, for instance, the logistic regression later on in the text. If this is the case, you do not need to indicate the abbreviation either

**Response:** Thank you very much for your constructive comment.

**After revised: (Page 2 Line 4-7)- Manuscript**

"The statistical method" Are you referring to a statistically-based method
We are referring to statistically-based methods, instead of statistical methods, and have been corrected throughout the revised manuscript.

**After revised: Lines 24 to 28, have been revised on Page 2 Line 4-7 – Manuscript**

The statistically-based method could be bivariate (Chung & Fabbri, 2012; Chung & Fabbri, 2003; Dai & Lee, 2002; Wubalem, 2020), a multivariate method (Atkinson & Massari, 1998; Polykretis et al., 2019), Artificial Neural Network (*ANN*) method (Ortiz & Martínez-Graña, 2018; Tsangaratos & Benardos, 2014; Vakhshoori et al., 2019) or Machine Learning Techniques (*MLTs*) (Tien Bui et al., 2012; Youssef & Pourghasemi, 2021; Kavzoglu et al., 2014).

**Comment 8:** Line 29: "The robustness of the statistical method is, however, suspect". Of course, there is no perfect model. But you cannot say that "The robustness of the statistical method is, however, suspect". Based on what? You can be critical for sure and mention some uncertainties, but not like this. Please revise the line.

**Response: Response:** Thank you very much for your constructive comment.
We agree with your comment. Following the reviewer's comments, the sentence has been deleted from the manuscript

**Comment 9:** Line 29: "statistical method generates landslide maps" What do you mean? Do you mean "landslide susceptibility maps"? You are resisting saying what we are really talking about. Susceptibility? Hazard? Or something else?

**Response: Response:** Thank you very much for your constructive comment.

We mean to write that," hazard maps generated by the statistically-based method" are obtained from a combination of maps generated by different control points, which are assumed to be independent of

each other. The sentence has been revised on Page 2 Lines 8 - 9. of the manuscript as,

**After revised: (Page 2 Lines 8 to 9) - Manuscript**

Hazard maps generated by some of these Statistically-based methods are obtained from a combination of maps generated using control points, whose predictive variables suffer from multicollinearity.

**Comment 10:** Lines 29-31: "Because the statistical method generates landslide maps by using a combination of maps generated by different control points that are assumed to be conditionally independent of each other, thus questioning its accuracy". What does this mean now? I have a hard time following the logic behind your argument. You are saying they? Then please say it clearly. Btw, I would say this is quite a minor issue among many others regarding the statistically-based method and actually, there are ways to deal with this issue in the literature. If you would like to be really critical, you should come up with better arguments/stronger. Also, are you sure that these papers support your argument (e.g., Youssef & Pourghasemi, 2021). Or these are the examples that you think that they are clearly representing the problem you mention.

**Response: Response:** Thank you very much for your constructive comment.
We agree with your comment. Following the reviewer's comments, the sentence has been revised on page 2 Lines 8-12 of the manuscript to read

**After revised: (Page 2 Lines 8-12) – Manuscript**

"statistically-based methods assume that landslide controlling factors are conditionally independent of each other (e.g., Youssef et al., 2016). Hazard maps generated by some of these Statistically-based methods are obtained from a combination of maps generated using control points, whose predictive variables suffer from multicollinearity. The multivariate statistically-based methods are also suitable for large and complex areas. However, the method's robustness highly depends on the database used for the analysis. And only conditionally identical to those in the database can be predicted (Tien Bui et al., 2012; H. Y. Tsai et al., 2019)".

We think multicollinearity among independent variables usually results in less reliable statistical inferences and, in this case, produces a hazard map that reflects the actual condition of a study area. Although using independent variables that are not correlated or repetitive when building multiple regression models that use two or more variables may solve this situation, an oversight could cause multicollinearity. This is why we see this as a drawback of the statistical method. Also, with the multivariate statistically-based methods, only conditionally identical data to those in the database can be predicted, meaning an unidentical database cannot be predicted, which also adds to the drawbacks of the statistical method.

**Comment 11:** Lines 31-32: "Recent engineering earthquake induced landslides and displacement analysis (Rathje et al., 1998; Jibson & Keefer, 1993; Saygili, 2008)" these do not sound "recent" to me. Btw, please rewrite the line, there is no such thing called "engineering earthquake induced landslides". Also, "analysis" should be "analyses".

**Response: Response:** Thank you very much for your constructive comment.

Recent was used in the sentence because the engineering method for landslide analyses is more contemporary compared to the statistically-based method. The sentence has been revised on page 2 lines 12-14 of the manuscript to read,

**After revised: (Page 2 Lines 12-14) – Manuscript**

"Engineering methods for earthquake-induced landslides and displacement analyses are done using the sliding block displacement method, which is a compromise in complexity between simple pseudo-static analysis and complex numerical simulation engineering methods."

**Comment 12:** Line 34: "Newark Rigid Dynamic Block Model" should be "Newmark's sliding block method". Please do the same corrections through the text.

**Response: Response:** Thank you very much for your constructive comment.

We agree with your comment. Following the reviewer's comments, the term 'Newmark Rigid Dynamic Block Model', has been corrected and rewritten as

"Newmark's sliding block method", throughout the manuscript.

**Comment 13:** Lines 34-36: "The precision of the Newark Rigid Dynamic Block Model cannot be misconstrued, as it produces a stronger correlation between the estimated sliding block displacement and the mapping location of the earthquake-triggered landslide". Please do not say "cannot be misconstrued" out of blue and please support your argument by citing the literature. This is quite a subjective statement I would say. As there is no perfect machine learning technique to assess the spatial distribution of landslides, the same is also valid for their physically-based counterparts. If you would like to list the pros and cons of both approaches, you have to do it in an objective manner. Btw, this may not be even required because I still do not understand where you want to go from here. Do we really need to list all these here? Are these relevant for this paper? There is a large literature associated with both approaches and no need to destroy or glorify one of them compared to the other. However, if you would like to do this then do it properly. For instance, how accurately can you identify geotechnical parameters to run a regional-scale landslide susceptibility/hazard analyses?

**Response: Response:** Thank you very much for your constructive comment.

We agree with your comment. Following the reviewer's comments, the comment has been revised on Page 2 Lines 17 -20 of the manuscript as below,

**After revised: (Page 2, Lines 17 -20) – Manuscript**

Newmark's sliding block method produces a stronger correlation between the estimated displacement and the mapping location of the earthquake-triggered landslide, making it a good engineering method suitable for predicting earthquake-induced landslides.

**Comment 14:** Line 38: Why is that the final stage?

**Response: Response:** Thank you very much for your constructive comment.

We agree with your comment. Following the reviewer's comments, the sentence "final stage" has been deleted from the manuscript.

**Comment 15:** Line 39: "achievable through slope mapping units". Please first tell us what the slope unit is and why you prefer working with it. Also, please prefer using either "Slope units" or "mapping units" not both

*Response: Response: Thank you very much for your constructive comment.*

The research has made two proposals, first is a displacement method that considers the effect of depth in its analysis. The proposed displacement method is being applied on a regional scale. Therefore, the area must first be divided into sampling units of landslide hazard zones in which every landslide influence factor can be allocated. And these landslide hazard zones are termed mapping units. The popular mapping units in landslide hazard assessment include grid cells, slope units, etc. (unit). According to hydrological theory, a "slope unit" is considered a watershed defined by ridge and valley lines and is used to divide spaces into smaller regions for easy analysis. Slope units are usually extracted from a digital elevation model (*DEM*) using geographic information system (*GIS*) software (Wang et al., 2019).

As Xie et al. (2003) highlight, slope units can represent natural slope topographic boundaries in their natural conditions because it uses naturally marked units of slope to represent natural landscape events. This makes the slope unit the best method for landslide hazard zonation. The research proposed a new slope unit extraction method to solve slope heterogeneity and boundary effects associated with the hydrological slope unit extraction method. (Note: The revised version of the manuscript has omitted the conventional method because its delineation is similar to the hydrological method).

**Comment 16:** Line 40: You started using susceptibility (e.g., in line 39) and hazard (e.g., in line 40) terms and which is ok but do not use them as if you can use them interchangeably. They are not the same thing, right?

**Response: Response:** Thank you very much for your constructive comment.

*We agree with your comment. Following the reviewer's comments, w*e admit that susceptibility and hazard maps aren't the same and cannot be used interchangeably in the manuscript. Susceptibility has been changed hazard throughout the manuscript (except places not applicable).

**Comment 17:** Line 42: "slope unit model" there is no model, just slope units. Also, if you are referring to slope units, it is more appropriate to cite papers that proposed slope units, not the ones that only used them based on available sources

**Response: Response:** Thank you very much for your constructive comment.

We agree with your comment. Following the reviewer's comments, we have realized that *both slope unit and grid cells are neither models nor methods, we have taken note and made the necessary corrections throughout the manuscript. Some of the publications cited in the manuscript propose slope units* (e.g. Cheng & Zhou, 2018; Wang et al., 2019, 2020; Tsai et al., 2019; Xie et al., 2003)

*After revised***:  References**

Cheng, L., & Zhou, B. (2018). A new slope unit extraction method based on an improved marked watershed. MATEC Web of Conferences, 232, 1–5. https://doi.org/10.1051/matecconf/201823204070

Wang, K., Zhang, S., DelgadoTéllez, R., & Wei, F. (2019). A new slope unit extraction method for regional landslide analysis based on morphological image analysis. Bulletin of Engineering Geology and the Environment, 78(6), 4139–4151. https://doi.org/10.1007/s10064-018-1389-0

Tsai, H. Y., Tsai, C. C., & Chang, W. C. (2019). Slope unit-based approach for assessing regional seismic landslide displacement for deep and shallow failure. Engineering Geology, 248(January 2018), 124–139. https://doi.org/10.1016/j.enggeo.2018.11.015

Xie, M., Esaki, T., Zhou, G., & Mitani, Y. (2003). Geographic Information Systems-Based Three-Dimensional Critical Slope Stability Analysis and Landslide Hazard Assessment. Journal of Geotechnical and Geoenvironmental Engineering, 129(12), 1109–1118. https://doi.org/10.1061/(ASCE)1090-0241(2003)129:12(1109)

**Comment 18:** Line 44: "(Xie et al., 2003)" please remove the parentheses

**Response: Response:** Thank you very much for your constructive comment.

We agree with your comment. Following the reviewer's comments, the parentheses have been removed on page 3 line 1. And the sentence reads

*After revised*: **(Page 3, Line 1-3) - Manuscript**

As highlighted by Xie et al., (2003), a limitation of the grid-cell mapping unit method is its inability to represent natural slope topographic boundaries in their natural condition because it uses artificially marked cells of a block to represent the natural landscape event

**Comment 19:** Line 48: "the best mapping unit for earthquake-induced landslide and displacement analysis" Do you mean it is the best mapping unit specifically for earthquake-induced landslides? Why is that? How about rainfall-triggered landslides?

**Response: Response:** Thank you very much for your constructive comment.

We used the sentence "the best mapping unit for earthquake-induced landslide and displacement analysis" precisely because this study focuses on earthquake-induced analysis, including external factors (for instance, soil properties). These external factors are also influenced by rainfall. As such, we thought it would be too bold to make such a statement. However, Following the reviewer's comments, the sentence has been revised on page 3, lines 4-6 of the manuscript to read,

*After revised*: **(Page 3, Line 4-6) – Manuscript**

According to hydrological theory, a "slope unit" is considered a watershed defined by the ridge and valley lines and is used to divide spaces into smaller regions for easy analysis, making the method more related to the geological environment, hence the best for landslide and displacement analysis.

**Comment 20:** Line 48: "The slope unit method" Slope unit is not the method but the output of some landscape partitioning methods which you haven't mentioned yet.

**Response: Response:** Thank you very much for your constructive comment. We agree with your comment.

The sentence "slope unit method" is a mistake on the part of the authors; we meant to write, "The slope unit extraction method proposed." However, the sentence has been removed from the revised manuscript.

**Comment 21:** Lines 51-54: Please separately cite the corresponding paper of each method you mention.

**Response:** Thank you very much for your constructive comment.

The corresponding paper for the hydrological slope unit extraction method has been accordingly cited in the revised manuscript (Mesut et al., 2011).

**After revised:** **References**

Mesut, T., & David, F., (2011). GeoRisk 2011. Delineation of slope profiles from Digital Elevation Models for landslide hazard analysis © ASCE 2011 403. GeoRisk 2011, 403–410. doi:10.1061/41183(418)87.

**Comment 22:** Lines 62-63: "The conventional watershed method for slope unit extraction" Please be specific and cite corresponding papers.

**Response: Response:** Thank you very much for your constructive comment.

The conventional slope unit extraction method involves delineating a catchment from DEM in ArcGIS software. The procedure is repeated for an invented DEM. The two catchments are merged to end the procedure for slope unit delineation. With the hydrological slope unit extraction method, the method for the extraction of the slope unit involves delineating a watershed from a DEM, then reversing the DEM to delineate another watershed. The two watersheds are merged to end the extraction of the slope unit.

The conventional slope unit extraction method follows the same trend as the hydrological slope unit extraction method but for minor details.

The conventional slope unit extraction method has been removed from the revised manuscript. And the hydrological slope unit extraction method has been cited in answer to question 24.

**Comment 23:** Line 68: "The application of the framework is validated in Ghana." Which earthquake is that?

**Response: Response:** Thank you very much for your constructive comment.

The application of the framework for the slope unit extraction method and the displacement of slopes is validated in Ghana.

Although Ghana does not have a strong landslide database, however, has had a series of earthquakes, as shown in the table below.

Using the proposed slope unit extraction and displacement methods, these earthquake histories (table below) were used alongside the geological parameters and soil material properties to predict Ghana's vulnerability to future landslide hazards.

**After revised:    Page 36 - Manuscript**

*Table 1* Earthquake Record of Ghana indicating earthquake parameters

| No. | Year | Magnitude ($M_l$) | Intensity ($I_n$) | Surf. Mag. ($M_s$) | Source |
|-----|------|-------------------|-------------------|---------------------|--------|
| 1 | 1615 | | | | Ambrasey's and Adams, 1986, NNA |
| 2 | 1636 | 5.7 | IX | North Axim | Ambrasey's and Adams, 1986, NNA |
| 3 | 1788 | 5.6 | | Accra | British Geological Survey, BGS |
| 4 | 1862 | 6.8 | IX | Accra | Ambrasey's and Adams, 1986, NNA |
| 5 | 1858 | 4.5 | | West of Accra | Ambrasey's and Adams, 1986, NNA |
| 6 | 1871 | 4.6 | VI | Accra | Ambrasey's and Adams, 1986, NNA |
| 7 | 1872 | 4.9 | VII | Accra | Ambrasey's and Adams, 1986, NNA |
| 8 | 1879 | 5.7 | | Accra | Ambrasey's and Adams, 1986, NNA |
| 9 | 1906 | 6.2 | VIII | Ho | Ambrasey's and Adams, 1986, NNA |
| 10 | 1907 | 5.0 | | Accra | Ambrasey's and Adams, 1986, NNA |
| 11 | 1939 | 6.5 | IX | Accra | Ambrasey's and Adams, 1986, NNA |
| 12 | 1948 | 4.0 | | Accra | Ambrasey's and Adams, 1986, NNA |
| 13 | 1964 | 4.7 | | Near Akosombo | Akoto and Anum, 1992, AKO |
| 14 | 1969 | 4.8 | | Offshore | United State Geological Survey, USG |
| 15 | 1997 | 4.7 | | Accra District | Internal Seismological centre, ISC |
| 16 | 1992 - 2002 | 1-3 | IV | Ho, Accra | Ambrasey's and Adams, 1986, NNA |
| 17 | 1615 | 4.0 | | Accra | Ambrasey's and Adams, 1986, NNA |

**Comment 24:** Line 70: "The impact of cohesion c is negligible, therefore neglected" In which context? You did not say anything about the landslides you examined. You can keep this statement for your method section and better explain it there

**Response: Response:** Thank you very much for your constructive comment.

*First, the study predicted the possibility of landslide hazards in Ghana using the country's current geological parameters, soil properties, topography, DEM, and earthquake histories. In the study* by Jibson & Keefer (1993)*, a* threshold displacement of 5-10 cm is deemed reasonable, and 5-10 cm indicates a failure. This study is used to determine possible landslide inventories in Ghana and is used to validate the study. *The manuscript initially neglected the effect of cohesion in the analysis. Still, upon further reflection, t*he rigid and flexible method has been modified to include cohesion as presented in the manuscript Page 7, (Lines 5-30)]. We initially ignored the effect of cohesion because, according to Biondi et al. (2007), consideration for the effect of pore water pressure during stability analysis of slope is usually done for cohesionless soil materials. As such, this study decided to base the displacement analysis on designing for the worse possible condition. Thus, neglecting the effect of cohesion. However, after further consideration based on the reviewer's comment, the study omitted the idea of neglecting the effect of cohesion in its analysis. And has included the effect of soil cohesion in the determination of the $F_s$, hence including the effect of cohesion in the displacement analysis as stated on Page 7 of the manuscript

**Comment 25:** Lines 71-72: I did not understand what you mean here.

**Response: Response:** Thank you very much for your constructive comment.

71-72 reads, " The paper also underlines the possibility of the proposed model for displacement analysis of shallow and deep slope failures considering the pore water pressure during the computation of the factor of safety $F_s$."

The general idea behind the study is to propose a new slope unit delineation method. The delineated slope unit extracted is used alongside Newmark's sliding block displacement method to predict landslide hazards in Ghana. Newmark's sliding block method is useful for the rapid prediction of a seismic-induced landslide. Still, it always considers the defined slope units to be infinite sets of grids, hence having definite depth (usually less than 3m).

However, Rathje & Antonakos, (2011) pointed out that ground motion parameters for displacement analysis on a regional scale can be altered to consider both shallow and deep failure due to their interaction with the sliding soil material. Therefore, this study decided to alter the sliding block method by Zhang et al. (2019) to overcome the problem of defining slope displacements as an infinite sliding block with shallow depth to consider it a finite failure slope with depth above 3m.

We realized that if the slope displacement is above 3m, then there is a need to consider the effect of underground water pressure when determining the factor of safety Fs (a measure for determining the slope's stability).

The $F_s$ method is altered from the regular one in Eq.1 proposed by Jibson & Keefer, (1993) to the adjusted one in Eq. 2, which is applicable for deep failure.

$$F_s = \frac{\tau_{ss}}{\tau_{st}} = \frac{c + \sigma \tan \varphi}{\tau} \qquad (2)$$

where $c$ is the cohesion, $\sigma$ is the effective stress, $\tau$ is the shear stress component parallel to the failure surface, and $\varphi$ is the rock's friction.

$$F_s = \frac{c + (\gamma - m\gamma_w)d\cos\beta\tan\varphi}{\gamma d\sin\beta}\left[1 - r_u\right] \qquad (5)$$

where m is the percentage of failure thickness saturated. Eq. (1) is primarily suitable for an infinite slope because the $F_s$ for infinite slopes are not dependent on the slope's depth d but rather the $c$, $\varphi$, and β. The approach could also be used to compute the $F_s$ for a finite slope by adding the effect of pore water pressure ($1 - r_u$) Eq. (2).

In this situation, the sentence means the slope displacement method can be used for finite and infinite grids depending on the $F_s$ method. *The sentence has been revised*

*After revised: (Page 14, Line 20-21) - Manuscript*

"The proposed displacement model applies to infinite and finite slope failures considering the effect of pore water pressure during the $F_s$ computation *for finite slopes*".

---

## Author Comment (AC4)

**Response letter to MS No: nhess-2022-43- Decision**

Entitled "Hazard Assessment of Earthquake –Induced Landslides Based on a Mechanical Slope Unit Extraction Method" 15th April.2022

Dear Editor.

We sincerely appreciate the editor's/reviewers time and effort in evaluating our manuscript. We agree with and accept all the comments and suggestions from the Editor and Reviewers. We have carefully and thoroughly revised the manuscript according to the editors/reviewer's questions and comments. In the revised manuscript, changes are shown by using the track changes mode. The point-to-point responses to the comments are detailed as follows:

**Editor**

**Comment:** You as the contact author are requested to individually respond to all referee comments (RCs) by posting final author comments (ACs) on behalf of all co-authors no later than 04 Aug 2022 (final response phase).

**Response:** Thank you for your comment and suggestion.

We agree with your opinion. According to the Reviewers' comments and suggestions, the questions have been answered based on the thoroughly revised manuscript.

**Reviewer (Dongliang Huang) (CC)#3**

This is a clear, concise, and well-written manuscript; the introduction is relevant and quite theory-based. The procedure and method follows a clear pattern.

The authors methodically dissect the fundamental viewpoint of the meanings of coseismic dislodging, presenting factors that record an infinite slope failure or flexible sliding block failure, as well as proposing a mechanical slope unit extraction method. However, few perspectives are found that need verification or correction.

**Comment 1:** The manuscript is not adequately organized in style and formatting.

**Response:** Thank you very much for your constructive comment. We agree with your comment. Following the reviewer's comments.

**After revised: The whole Manuscript**

The Manuscript is now adequately formatted by rearranging and restructuring the sentences and figures to suit the Journal's standard. Reference to figures and tables are very consecutive, and captions are informative and self-standing.

**Comment 2:** The introduction must adequately present the proposed slope unit method and the gap in the displacement method used.

**Response: Thank you very much for your constructive comment.**

We agree with your comment. Following the reviewer's comments, the introduction has been overhauled to have a balance on the literature and empirical relations available for the calculation of the coseismic displacement and the different solutions for landslide stability analysis, as written below.

**After revised: Introduction Page 1 (Line 18-24) and Page 1, (Line 1-35) -Manuscript**

Earthquakes are the most dangerous natural hazards, posing the most significant risk to life and property. Since the 1980s, earthquakes-induced landslides have caused many deaths and economic losses. For example, the Chi-Chi earthquake-triggered about 9000 landslides (Tsai et al., 2000), and the Guatemala earthquake triggered 10,000 landslides

(Hamilton, 1997). Landslides mainly occur when acting forces exceed the strength of earth materials that composes a slope, and its evaluation provides general insight into future earthquake-induced landslides based on medium and long-term predictions of earthquake distribution to provide a possible mitigation measure to control its impact on life and properties (Bray et al., 2018; Salunkhe et al., 2017; Tsai & Chien, 2016; Wang & Lin, 2010; Zhang et al., 2019). Statistically-based methods based on historical landslide distribution were typically used in the early days for hazard zonation. However, the engineering approach (i.e., physically-based modeling) with the application of slope stability analysis models has recently been intensively studied and used to analyze landslides (Cencetti & Conversini, 2003; Tsai et al., 2019).

The statistically-based method could be bivariate (Chung & Fabbri, 2012; Chung & Fabbri, 2003; Dai & Lee, 2002; Wubalem, 2020), a multivariate method (Atkinson & Massari, 1998; Polykretis et al., 2019), Artificial Neural Network (ANN) method (Ortiz & Martínez-Graña, 2018; Tsangaratos & Benardos, 2014; Vakhshoori et al., 2019) or Machine Learning Techniques (MLTs) (Tien Bui et al., 2012; Youssef & Pourghasemi, 2021; Kavzoglu et al., 2014). Statistically-based methods assume that landslide controlling factors are conditionally independent of each other (e.g., Youssef et al., 2016). Hazard maps generated by some of these Statistically-based methods are obtained from a combination of maps generated using control points, whose predictive variables suffer from multicollinearity. The multivariate statistically-based methods are also suitable for large and complex areas. However, the method's robustness highly depends on the database used for the analysis. And only conditionally identical to those in the database can be predicted (Tien Bui et al., 2012; H. Y. Tsai et al., 2019). Engineering methods for earthquake-induced landslides and displacement analyses are done using the sliding block displacement method, which is a compromise in complexity between simple pseudo-static analysis and complex numerical simulation engineering methods (Ellen et al., 1998; Jibson & Keefer,

1993; Jibson et al., 2000; Saygili, 2008; Tsai et al., 2019; Wang et al., 2017; Shinoda & Miyata, 2017, Zhang et al., 2021). The sliding block method considers the landslide as a rigid non-deformable plastic body that slides on a plane of continuous dip and friction and only accurately predicts the displacement of an individual sliding body (Jibson, 2011; Tsai et al., 2019). The Newmark's sliding block method produces a stronger correlation between the estimated displacement and the mapping location of the earthquake-triggered landslide, making it a good engineering method suitable for predicting earthquake-induced landslides (Rathje et al., 1998; Tsai et al., 2019; Xie et al., 2003; Zhang et al., 2019). Newmark's sliding block method is the first for seismic-induced landslides' displacement analysis (Jibson, 2011; Newmark, 1965). The Newmark's sliding block method is useful for rapidly predicting a seismic-induced landslide by first diving the study area into numerous grids, especially during regional displacement assessment (Tsai et al., 2019). These grids are assumed to be infinite, hence having definite depth (usually less than 3m) and as well neglect slope geometry in their analysis therefore modelled asrigid blocks (Ellen et al., 1998; Jibson et al., 2000; Xie et al., 2003; Zhang et al., 2019). Rathje & Antonakos, (2011) pointed out that ground motion parameters for displacement analysis on a regional scale can be altered to consider both shallow and deep failure due to their interaction with the sliding soil material. A framework for predicting earthquake-induced displacement is therefore proposed based on Zhang et al., (2019) to overcome the problems of defining slope displacements as an infinite sliding block with shallow depth, to consider it as a finite failure, especially in less cohesive soil materials whiles considering the effect of pore pressure and slope geometry in determining the safety factor  $F_s$  to predict an unbiased  $k_v(g)$  for the displacement analysis. This framework is based on regional hazard analyses; therefore, the study area must first be divided into sampling units of landslide hazard zones in which every landslide influence factor can be allocated. These landslide hazard zones are mapping units (Ba et al., 2018).

Most popular Mapping units for earthquake-induced landslide displacement analysis include the grid-cell, slope unit, etc. (Schlögel et al., 2018; Tsai et al., 2019; Yu & Chen, 2020). Grid cells are regular square cells with a given size for the unit mapping of landslides and are not closely related to geological environments (Guzzetti et al., 1995). As highlighted by Xie et al., (2003), a limitation of the grid-cell is its inability to represent natural slope topographic boundaries in their natural condition because it uses artificially marked cells of a block to represent the natural landscape event.

According to hydrological theory, a "slope unit" is considered a watershed defined by the ridge and valley lines and is used to divide spaces into smaller regions for easy analysis, making the method more related to the geological environment, hence the best for landslide and displacement analysis (Guzzetti et al., 1995; Wang et al., 2017). Slope unit is more applicable than the grid-cell method because landslides occur on slopes; therefore, the slope unit represents the topographic feature more thoroughly than the grid cell (Wang et al., 2017; Xie et al., 2003). Slope units are usually extracted from the digital elevation model (*DEM*) using geographic information system (*GIS*) software (Wang et al., 2019). The method for the extraction of the slope unit involves delineating a watershed from a

DEM, then reversing the *DEM* to delineate another watershed. The two watersheds are merged to end the extraction of the slope unit (Mesut et al., 2011). This slope unit extraction method is termed the hydrological slope unit extraction method (Fig. 1a)(Mesut et al., 2011). Slope units extracted with the hydrological method are usually based on the surface hydrological process, making it impossible to identify variations in slope gradient beyond the hydrological flow direction, resulting in a sudden change in slope gradient. As such, slope units extracted using the hydrological methods suffer a heterogeneity effect primarily associated with slope units extracted using high-resolution DEM (Guzzetti et al., 1995; Wang et al., 2019, 2020). The hydrological slope unit extraction method again produces irregular boundaries and conjoined slope conditions. This occurs because it barely distinguishes inclined and horizontal planes of deep valleys and high mountainous terrains (Wang et al., 2019). Tedious manual post-extraction corrections are needed to make the slope unit acceptable (Cheng & Zhou, 2018; Wang et al., 2020).

This research proposes a new slope unit extraction method using GIS software for the earthquake-induced displacement framework analysis. The method combines catchment points, hydrological slope unit extraction method, and segmentation to overcome the limitations of the hydrological slope unit extraction method. The application of the slope unit extraction method and displacement framework is validated in Ghana. The prediction result of the slope unit extraction method is compared with the hydrological method. The displacement framework is also compared with the displacement method by Jibson et al., (2000); Tsai et al., (2019). The paper also underlines the possibility of the proposed model for displacement analysis of shallow and deep slope failures considering the pore water pressure during the computation of the factor of safety  $F_s$ .

**Comment 3:** Eliminating the boundary and heterogeneity effect of the hydrological slope unit extraction method are the main innovations of the newly proposed slope unit extraction method in the manuscript; however, they have not been adequately explained in the manuscript. An explanation of the heterogeneity and boundary effect will sufficiently improve the manuscript.

**Response: Thank you very much for your constructive comment.**

We agree with your comment. Following the reviewer's comments, We have overhauled the article, restructured it, and added some sentences and figures to enhance clarity on the heterogeneity defect in Fig.4, page 23, and Page 12, (Line 6-16) and 22 of the manuscript and presented it as stated below.

**After revised: 1) Slope unit heterogeneity [Fig.4, page 23 and Page 12, (Line 6-16) – Manuscript**

Slope angle is a critical factor for consideration in a landslide analysis (Wang et al., 2019). Fig.4 (c) shows the terrain profile line A'-A (convex area) with two flat slopes of 30 and 10 degrees, respectively, and a slope toe angle of 45. Fig.4 (a) illustrates a slope unit extracted using the point segmentation method showing line A'-A area with three local slope unit regions. Fig.4 (b) is a slope unit map

obtained using the hydrological method showing line A'-A (convex area) with two local slope unit regions and two angles. ArcGIS statistical tool is used to compute the slope angle of the slope unit region and profile in Fig.4 (a) and Fig.4 (b). Fig.11 (a) shows the terrain profile of the slope unit region determined using the hydrological method; the terrain shows just two slope angles, with the toe having a 45 angle and a second slope angle of 10 instead of 30 (as was demonstrated in Fig.4 (c)). This indicates that the hydrological method underestimates the slope angles compared to the point segmentation method. The point segmentation method divided the area into three slope units and three angles having the exact sizes and conforming homogenously to the terrain in Fig.4 (c).